# Faster Generic Identification in Tree-Shaped Structural Causal Models

**Yasmine Briefs**
Max-Planck-Institute for Informatics
Saarland Informatics Campus
Saarbrücken, Germany
ybriefs@mpi-inf.mpg.de

**Markus Bläser**
Saarland University
Saarland Informatics Campus
Saarbrücken, Germany
mblaeser@cs.uni-saarland.de

## Abstract

Linear structural causal models (SCMs) are used to analyze the relationships between random variables. Directed edges represent direct causal effects and bidirected edges represent hidden confounders. Generically identifying the causal parameters from observed correlations between the random variables is an open problem in causality. Gupta and Bläser (AAAI 2024, pp. 20404–20411) solve the case of SCMs in which the directed edges form a tree by giving a randomized polynomial time algorithm with running time $O(n^6)$. We present an improved algorithm with running time $O(n^3 \log^2 n)$ and demonstrate its feasibility by providing an implementation that outperforms existing state-of-the-art implementations.

## 1 Introduction

Understanding and predicting causal effects—while distinguishing them from mere statistical correlations—is a key objective in empirical sciences. For instance, uncovering the causes of diseases and health outcomes is essential for developing effective prevention and treatment strategies. A widely accepted method for establishing causality is the randomized controlled trial (RCT). However, RCTs require direct intervention in the system under study, which is not always feasible due to ethical, financial, or technical constraints. For example, to study the long-term effects

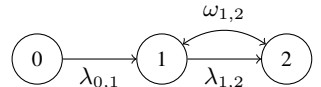

Figure 1: An IV example. 0 is e.g. the tax rate on tobacco, 1 the number of cigarettes smoked, and 2 whether one developed lung cancer.

of air pollution on human health, it would be unethical and impractical to expose individuals to harmful pollutants intentionally. In such cases, researchers turn to alternative methods that infer causal relationships by leveraging observational data in conjunction with prior knowledge about the system's structure. This challenge, known as the problem of identification in causal inference, is a central concern in many fields, including modern ML.

A central component of the causal inference framework is how the underlying structure captures the true generative mechanism of the system. This is typically formalized using *structural causal models* (SCMs) [Pearl, 2009, Bareinboim et al., 2022]. Here, we focus on the problem of *identification* within the class of *linear* SCMs, also known as *linear structural equation models* (SEMs) [Bollen, 1989, Duncan, 1975]. These models represent causal relationships among observed random variables by assuming that each variable $X_j$ is a linear combination of other variables and an unobserved noise term $\varepsilon_j$, where the noise terms are jointly normally distributed with zero mean: $X_j = \sum_i \lambda_{i,j} X_i + \varepsilon_j$. The error terms are characterized by a covariance matrix $\Omega = (\omega_{i,j})$, which models hidden confounding terms. Linear SCMs can be represented as a graph with nodes $\{0, \ldots, n-1\}$ corresponding to the $n$ random variables. There are two types of edges: A directed edge $i \to j$ represents a linear influence $\lambda_{i,j}$ of a parent node $i$ on its child $j$. A bidirected edge $i \leftrightarrow j$ represents a correlation $\omega_{i,j}$ between

39th Conference on Neural Information Processing Systems (NeurIPS 2025).

error terms $\varepsilon_i$ and $\varepsilon_j$. Figure 1 shows a classical example of an SCM. In this paper, we primarily consider *recursive* models, where the directed structure is acyclic—i.e., $\lambda_{i,j} = 0$ for all $i > j$. Let $\Lambda = (\lambda_{i,j})$ denote the adjacency matrix of all *directed edges*, and $\Omega = (\omega_{i,j})$ the adjacency matrix corresponding to *bidirected edges* (i.e., covariances of the error terms). Then the covariance matrix $\Sigma = (\sigma_{i,j})$ of the observed variables $X_0, \ldots, X_{n-1}$ is given by: $\Sigma = (I - \Lambda)^{-T} \Omega (I - \Lambda)^{-1}$, where $I$ is the identity matrix [Foygel et al., 2012]. The central challenge lies in *recovering $\Lambda$ from the observed covariance matrix $\Sigma$*, under the assumption that $\Omega$ is unknown. Once $\Lambda$ is identified, we can also recover $\Omega$ using the equation above.

Most approaches for identification in practice are based on *instrumental variables (IV)*, in which the causal direct effect is identified as a fraction of two covariances [Bowden and Turkington, 1990]. For example in Figure 1, one can calculate first $\omega_{0,0}\lambda_{0,1} = \sigma_{0,1}$ and then $\lambda_{1,2} = \frac{\omega_{0,0}\lambda_{0,1}\lambda_{1,2}}{\omega_{0,0}\lambda_{0,1}} = \frac{\sigma_{0,2}}{\sigma_{0,1}}$. (This follows from Wright's trek rule, see Section 2.) The variable $X_0$ is then called an instrument. This method is *sound*, that is, when it identifies a parameter, then it is always correct. But it is not *complete*, that is, it might fail to identify a parameter by IVs, but the parameter might be identifiable by other means.

An SCM is *globally identifiable* when we can *always* uniquely recover the parameters $\Lambda$ given observed covariances $\Sigma$. Global identification can be decided in polynomial time [Drton et al., 2011, Thm 2]. However, it is a very strong property. In Figure 1, we can recover the parameter $\lambda_{1,2}$ as $\frac{\sigma_{0,2}}{\sigma_{0,1}}$. If $\sigma_{0,1} = 0$, then the identification fails, so the SCM is not globally identifiable. But identification fails only in the degenerate case that $\sigma_{0,1} = 0$. This leads to the concept of *generic identifiability*.

**Definition 1** (Generic identification). *A parameter $\lambda_{p,i}$ is generically identifiable if for almost all $\Sigma$ that can be generated by the model, there is only one solution for $\lambda_{p,i}$.*

"Almost all" above means that the $\Sigma$ for which identification fails form an algebraic variety or alternatively and equivalently, a set of measure $0$. A weaker notion is *generically $k$-identifiable*: Here we ask whether there are at most $k$ solutions for the parameter $\lambda_{p,i}$ for almost all $\Sigma$. If such a parameter $\lambda_{p,i}$ is generically identifiable, known algorithms also provide a symbolic expression to calculate the coefficient. For instance, in Figure 1, the parameter $\lambda_{1,2}$ is given by the rational expression $\frac{\sigma_{0,2}}{\sigma_{0,1}}$. As described above, this is undefined when $\sigma_{0,1} = 0$, however, the equation $\sigma_{0,1} = 0$ specifies an algebraic variety. Therefore, $\lambda_{1,2}$ is generically identifiable. Characterizing all linear structural causal models that are generically identifiable is a major open problem in causality, see e.g. [Foygel et al., 2012, Problem 1].

## 1.1 Previous results

Conditional instrumental variables (cIVs) represent a natural extension of simple IVs [Bowden and Turkington, 1990, Pearl, 2001], with an associated identification method that admits an efficient, polynomial-time algorithm for discovering valid cIVs [van der Zander et al., 2015]. Several more expressive identification criteria have been introduced, each supported by polynomial-time algorithms. These include instrumental sets (IS) [Brito and Pearl, 2002a], the half-trek criterion (HTC) [Foygel et al., 2012], instrumental cutsets (ICs) [Kumor et al., 2019], and auxiliary instrumental variables (aIVs) [Chen et al., 2015]. Further generalizations, such as the generalized half-trek criterion (gHTC) [Chen, 2016, Weihs et al., 2018] and auxiliary variables (AVS) [Chen, 2016, Chen et al., 2017], are also tractable, assuming the in-degree of each node in the causal graph is bounded. The auxiliary cutsets (ACID) algorithm [Kumor et al., 2020] unifies these approaches by subsuming all instances identifiable by the aforementioned methods.

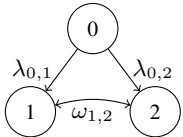

Figure 2: An example from phylogenetics. The variables model a specific trait of the species. Species 0 is an ancestor of 1 and 2 and we have a direct effect. The bidirected edge between 1 and 2 can for instance model that 1 and 2 live in a similar environment.

All methods mentioned above are sound. However, none of them is complete, that is, there are examples in which they fail to identify parameters that are identifiable in principle. There are complete methods based on Gröbner bases [García-Puente et al., 2010]. Recently, Dörfler et al. [2024] proved that generic identification can be expressed in the theory of the reals and can therefore be decided in PSPACE (and with singly exponential running time). In practice, these algorithms however can only identify SCMs with only few nodes. Van der Zander et al. [2022] report that a Gröbner basis based algorithm needed 4 months to solve 879 sample SCMs with 8 nodes each (see

Section 5.1). While there is no hardness result known for generic identification, we suspect that it is a hard problem. Dörfler et al. [2024] proved that a related notion of identification, called numerical identification, is hard for the universal theory of the reals (which implies coNP-hardness).

When there is little hope for an efficient algorithm for general SCMs, it is natural to look at restricted classes of SCMs. Van der Zander et al. [2022] introduced *tree-shaped SCMs*, in which the directed edges form a tree with all edges pointing away from the root. There are no restrictions on the bidirected edges. Figure 1 shows a tree-shaped SCM, there $0 \to 1 \to 2$ is even a path. Tree-shaped SCMs naturally occur when we have hierarchical structures, like in biology. Figure 2 presents a toy example from phylogenetics, see e.g. [Thorson et al., 2023] for much larger examples. Van der Zander et al. [2022] give a PSPACE-algorithm called *treeID* for tree-shaped SCMs, which at that time was an improvement, since all known polynomial time methods for generic identification perform poorly on this class of SCMs [van der Zander et al., 2022, Prop. 2], so one had to resort to Gröbner bases. TreeID is complete for the class of tree-shaped SCMs, which follows from the later work by Gupta and Bläser [2024a]. In this work, Gupta and Bläser [2024a] give a randomized polynomial-time algorithm that is complete for tree-shaped SCMs. However, their running time is $O(n^6)$, which is feasible only for small instances.

The treeID algorithm by van der Zander et al. [2022] has been implemented, it is available in the *DAGitty* package [Textor et al., 2016]. The *SEMID* package [Barber et al., 2023] is a state-of-the-art R-package for generic identification in SCMs.

## 1.2 Our results

As our first main contribution, we present a randomized algorithm that is complete for tree-shaped SCMs with provable running time $O(n^3 \log^2 n)$. Our key insight is an improved polynomial identity testing scheme that we use to implement several of the steps in the algorithm of Gupta and Bläser [2024a] significantly faster. As our second contribution, we provide a C++ implementation of our algorithm. We compare it with two state-of-the-art implementations for generic identification, treeID of the DAGitty package, which has been explicitly designed for tree-shaped SCM, and the established HTC criterion of the SEMID R-package, which is a general purpose criterion, but which is not complete, even not for tree-shaped SCMs.

Our algorithm is randomized. It is complete in the strongest possible sense. For any tree-shaped SCM, it gives the correct answer with high probability, where the probability is taken over the internal randomness of the algorithm. For instance for SCMs with $n = 200$ nodes, the answer is correct with probability $1 - 4.1 \cdot 10^{-6} \geq 0.999995$, see Appendix C for calculations. This is a mathematically proven worst case guarantee. The concrete bound comes from the fact that we do all computations in built-in integers, which speeds up the implementation a lot. But by running the algorithm several times, the success probability goes to $1$ exponentially fast by the Chernoff bound, see e.g. [Mitzenmacher and Upfal, 2005].

The paper is structured as follows: In the next section, we give more details on generic identification. In Section 3, we describe the algorithm by Gupta and Bläser [2024a], since we follow its general blueprint. However, we are able to speed up all of their steps considerably. Section 4.1 exemplifies this for rank-computation, which we speed up to $O(n^2)$ compared to $O(n^3 \log n)$. The key construction is an improved identity testing algorithm, presented in Section 4.2. Although generic identification amounts to deciding whether a polynomial system over the reals has one or more solutions, by using identity testing, we can reduce this to integer computations instead of manipulation of formal polynomials, which is crucial for achieving polynomial running time. In Section 4.3, we describe how to speed-up further parts of the algorithm by Gupta and Bläser. Due to space limitations, we only give a sketch, the full explanations along with proofs can be found in Appendices B.1, B.2, and B.3. In Section 5, we compare the running time of our implementation with treeID and HTC/SEMID. Appendix A contains proofs omitted due to space constraints. Appendix C contains concrete estimates for the error probabilities of our implementation.

## 2 Generic identification

A *mixed graph* $M = (V, D, B)$ is a graph with node set $V$ and two types of edges, directed and bidirected ones. We can assume w.l.o.g. that the node set $V = \{0, 1, \ldots, n-1\}$. $D$ is the set of

directed edges. We denote directed edges either by $(i, j)$ or $i \to j$. We will always assume in this work that the graph $(V, D)$ consisting of the directed edges is acyclic, that is, the model is recursive. Furthermore, the nodes are topologically sorted w.l.o.g., that is, for every directed edge $i \to j$, we have $i < j$. $B$ is the set of bidirected edges, which we denote by $\{i, j\}$ or $i \leftrightarrow j$. Each node corresponds to a random variable $V_i$. With each directed edge $i \to j$, we associate an indeterminate $\lambda_{i,j}$ and with every bidirected edge $u \leftrightarrow v$ an indeterminate $\omega_{u,v}$. The random variable $V_i$ depends linearly on its parents and an additional error term $\epsilon_i$, that is, we can write $V_i = \sum_j \lambda_{j,i} V_j + \epsilon_i$ where the sum runs over all $j$ such that $j \to i$ is a directed edge in $D$. The error terms are normally distributed and a bidirected edge $i \leftrightarrow j$ indicates that the error terms are potentially not independent and the covariance between $\epsilon_i$ and $\epsilon_j$ is given by the parameter $\omega_{i,j}$. This is a standard model, also known as *linear structural equation model*, see [Drton, 2018, Pearl, 2009].

Let $\sigma_{i,j}$ denote the "observed" covariance between the random variables $V_i$ and $V_j$. Wright's trek rule relates $\sigma_{i,j}$ to the parameters of the mixed graph. A trek $\tau$ from $i$ to $j$ consists of two directed paths $i_0 \to i_1 \to \cdots \to i_k = i$ and $j_0 \to j_1 \to \cdots \to j_\ell = j$ in $(V, D)$ such that either $i_0 = j_0$ or there is a bidirected edge $i_0 \leftrightarrow j_0 \in B$. The monomial $M(\tau)$ of $\tau$ is $\omega_{i_0, j_0} \prod_{\kappa=0}^{k-1} \lambda_{i_\kappa, i_{\kappa+1}} \prod_{\mu=0}^{\ell-1} \lambda_{j_\mu, j_{\mu+1}}$. Now *Wright's trek rule states* that $\sigma_{i,j} = \sum_\tau M(\tau)$, where the sum is taken over all treks from $i$ to $j$.

Let $\Lambda = (\lambda_{i,j})$ and $\Omega = (\omega_{i,j})$ be the $n \times n$-matrices corresponding to the directed and bidirected edges. Since $V$ is topologically sorted, $\Lambda$ is upper triangular. The matrix $\Omega$ is symmetric. On the diagonals of $\Omega$, we will write the variances of the error terms $\epsilon_i$. We denote these variances by $\omega_{i,i}$, using double indices to make it compatible with the matrix notation. Let $\Sigma = (\sigma_{i,j})$ be the matrix containing the covariances between any two of the random variables $V_0, \ldots, V_{n-1}$. $\Sigma$ models the observed covariances and $\Lambda$ and $\Omega$ model the internal parameters that we want to identify. The $\sigma_{i,j}$ are polynomials in the parameters $\lambda_{i,j}$ and $\omega_{i,j}$, given by the matrix equation

$$\Sigma = (I - \Lambda)^{-T} \Omega (I - \Lambda)^{-1} \quad (1) \qquad \text{or equivalently} \qquad \Omega = (I - \Lambda)^T \Sigma (I - \Lambda), \quad (2)$$

see e.g. [Drton, 2018, (2.2)]. This is nothing but Wright's trek rule written in matrix form.

We call a bidirected edge $i \leftrightarrow j$ *missing* if it is not present in our mixed graph $M$, that is, $i \leftrightarrow j \notin B$. If $i \leftrightarrow j$ is missing, then the corresponding entry in $\Omega$ is $0$. It turns out that exactly the equations in (2) that correspond to these entries are useful for identification, see e.g. [Drton, 2018, Lemma 8.1]. (Intuitively this is clear, since if the edge is not missing, then $\Omega$ contains the variable $\omega_{i,j}$ in this position and since the variable only appears once in (2), we can set it in such a way that the equation is satisfied.)

In the case of a tree-shaped SCM with root $0$, these equations simplify: Van der Zander et al. [2022, Lemma 5] prove that a particular parameter $\lambda_{u,v}$ is generically identifiable (or in general $k$-identifiable) iff the system of equations

$$\lambda_{p,i} \lambda_{q,j} \sigma_{p,q} - \lambda_{p,i} \sigma_{p,j} - \lambda_{q,j} \sigma_{i,q} + \sigma_{i,j} = 0 \quad \text{for all } i \leftrightarrow j \notin B, i, j \neq 0, \quad (3)$$

$$\lambda_{p,i} \sigma_{0,p} - \sigma_{0,i} = 0 \quad \text{for all } 0 \leftrightarrow i \notin B \quad (4)$$

has a unique solution for $\lambda_{u,v}$ ($k$ solutions, respectively). In each equation, $p$ denotes the unique parent of $i$ and $q$ the unique parent of $j$ in the tree $(V, D)$. We will use this convention frequently in the following. If there is a missing edge $0 \leftrightarrow i$, then $\lambda_{p,i}$ is generically identifiable and even rationally identifiable as $\lambda_{p,i} = \sigma_{0,i}/\sigma_{0,p}$ by (4). So missing edges to the root allow for easy identification.

## 3 The algorithm by Gupta and Bläser

Gupta and Bläser [2024a] give a randomized polynomial-time algorithm for generic identification in tree-shaped SCMs. This algorithm is complete, that is, every parameter $\lambda_{p,i}$ is correctly declared as either generically identifiable, 2-identifiable, or unidentifiable (no other cases can occur). Their paper is mainly a contribution to algorithm theory, an algorithm with proven polynomial running time and correctness. The actual running time is $O(n^6)$ [Gupta and Bläser, 2024b]. As our two main contributions, we first improve the theoretical running time significantly to $O(n^3 \log^2 n)$ and second prove its practical feasibility by providing an implementation of our algorithm, which outperforms state-of-the-art implementations.

The key observation by Gupta and Bläser [2024a] is that (3) can be rewritten as

$$\lambda_{p,i} = \frac{\lambda_{q,j} \sigma_{i,q} - \sigma_{i,j}}{\lambda_{q,j} \sigma_{p,q} - \sigma_{p,j}} \qquad \text{or} \qquad \lambda_{q,j} = \frac{\lambda_{p,i} \sigma_{p,j} - \sigma_{i,j}}{\lambda_{p,i} \sigma_{p,q} - \sigma_{i,q}}. \quad (5)$$

This transforms $\lambda_{q,j}$ into $\lambda_{p,i}$ by the so-called *Möbius transformation*, see [Krantz, 1999], with the matrix $\left( \begin{smallmatrix} \sigma_{i,q} & -\sigma_{i,j} \\ \sigma_{p,q} & -\sigma_{p,j} \end{smallmatrix} \right)$ and $\lambda_{p,i}$ into $\lambda_{q,j}$ by the inverse Möbius transformation given by the negative of the adjoint matrix $\left( \begin{smallmatrix} \sigma_{p,j} & -\sigma_{i,j} \\ \sigma_{p,q} & -\sigma_{i,q} \end{smallmatrix} \right)$. If there is another missing edge, say $i \leftrightarrow k$, and let $(r, k)$ be the (unique) directed edge entering $k$, then we can express $\lambda_{r,k}$ in terms of $\lambda_{p,i}$ similar to (5). Since we can express $\lambda_{p,i}$ in terms of $\lambda_{q,j}$ by (5), we can express $\lambda_{r,k}$ in terms of $\lambda_{q,j}$. This is again done by a Möbius transformation and the matrix of this transformation is the product of the two matrices, cf. [Gupta and Bläser, 2024a, Eq. (6)].

If we can find a cycle of missing bidirected edges, that is, a directed cycle in the graph $(V, \bar{B})$, then we get an equation that contains only one variable of the form $\lambda_{p,i} = \frac{\lambda_{p,i} b + d}{\lambda_{p,i} a + c}$, which we can use to identify the corresponding parameter by solving for $\lambda_{p,i}$. Here the matrix $\left( \begin{smallmatrix} b & d \\ a & c \end{smallmatrix} \right)$ is the product of the matrices along the cycle of missing edges starting in the head $i$ of the directed edge $p \to i$ with label $\lambda_{p,i}$. The entries $a$, $b$, $c$, and $d$ are polynomials in the entries of $\Sigma$, which are itself polynomials in the entries of $\Lambda$ and $\Omega$. If $a = 0$ and $c - b \neq 0$, then the solution is given by the rational expression $\lambda_{p,i} = d/(c - b)$. If $a \neq 0$, then there are two solutions given by the symbolic expressions $\frac{-(c-b) \pm \sqrt{\Delta}}{2a}$, where $\Delta := (c - b)^2 + 4ad$ denotes the discriminant. In these two cases, we call the cycle *identifying*. However, it can happen that this equation is trivial. In this case $a = d = c - b = 0$, that is, the matrix $\left( \begin{smallmatrix} b & d \\ a & c \end{smallmatrix} \right)$ is a polynomial multiple of the $2 \times 2$-identity matrix [Gupta and Bläser, 2024a, Lemma 8].

So the task is to find a cycle of missing bidirected edges that is identifying. However, there are potentially exponentially many missing cycles. Van der Zander et al. [2022] deal with this issue by simply enumerating all cycles in the graph. Gupta and Bläser [2024a] design an algorithm to find a cycle that gives a nontrivial equation, if one exists, in polynomial time. Then the solution found can be transferred to the other parameters using the Möbius transformation on the missing edges.

The actual situation is more complicated, since a Möbius transformation is only a bijection if the corresponding matrix has full rank two. We call such a missing bidirected edge a missing rank-two edge. Gupta and Bläser [2024a, Lemma 25] show that whenever there is a missing rank-one edge $i \leftrightarrow j$ then there is also a missing bidirected edge between $i$ or $j$ and the root $0$ and therefore, we can identify $i$ or $j$ using (4). Algorithm 1 summarizes the algorithm by Gupta and Bläser [2024a]. We will follow this blueprint, but we will provide significantly improved realizations of its steps. To speed up Algorithm 1 it is sufficient to improve the identification in the rank-two components $C_1, \ldots, C_m$. However, we will also speed up the first step of the algorithm, the rank test, since it is very instructive and explains our new ideas. To identity the edges of the rank-two components, Algorithm 1 performs four steps:

**Detecting an identifying cycle:** This finds the length $t$ of a shortest identifying cycle or reports that none exists. (Line 7 in Algorithm 1)

**Finding an identifying cycle:** The procedure in the first step only detects whether there is an identifying cycle and computes its length. If there is one, then we find one using a technique called self-reduction. (Still line 7)

**Propagating the solutions:** The solution we get from the parameter identified by the cycle is either of the form $\lambda_{p,i} = d/(c - b)$ (one solution, rational expression, line 9) or $\lambda_{p,i} = \frac{-(c-b) \pm \sqrt{\Delta}}{2a}$ (two solutions, line 10). These solutions are now propagated to the other nodes in the current component using the Möbius transforms on the missing edges (which have all rank two). Rational expressions stay rational expression. In the case of two solutions, the terms become *fractional affine square-root terms of polynomials (FASTP)*, as defined by van der Zander et al. [2022]. A FASTP is an expression of the form $\frac{u + v\sqrt{\Delta}}{r + t\sqrt{\Delta}}$, where $u, v, r, t, \Delta$ are polynomials given by arithmetic circuits[1]. Note that the original expression for $\lambda_{p,i}$ is also a FASTP (with $t = 0$).

---

[1] An arithmetic circuit is a directed acyclic graph. The nodes with indegree 0 (input gates) are labeled with variables or constants from a given field. The other nodes are either labeled with $+$ (addition gate) or $*$ (multiplication gate). Nodes with outdegree 0 are output gates. An arithmetic circuit computes polynomials at the output gates in the natural way. Such arithmetic circuits occur in our setting for instance as iterated matrix products or determinants with polynomials as entries. See [Shpilka and Yehudayoff, 2010] for more details.

---

**Algorithm 1** Identification in tree-shaped SCMs

---

**Input**: A tree-shaped mixed graph $M = (V, D, B)$
**Output**: For each $\lambda_{p,i}$, we output whether it is generically identifiable, 2-identifiable, or unidentifiable. In the first two cases, we output corresponding FASTPs.
 1: Find all rank-1 edges in the missing edge graph.
 2: Let $C_1, \ldots, C_m$ be the connected components formed by the rank-two edges.
 3: **for** each connected component $C_i$ **do**
 4:    **if** $C_i$ contains a node connected to the root with a missing edge **then**
 5:       Propagate the result to all nodes in $C_i$ and produce corresponding rational expressions.
 6:    **else**
 7:       Find an identifying cycle in $C_i$ of minimum length $t$.
 8:       If no such cycle is found, report that all nodes of $C_i$ are unidentifiable.
 9:       If the cycle produces one solution, then propagate it to all the nodes of $C_i$ and compute corresponding rational expressions.
10:       If the cycle produces two solutions, then propagate them to all the nodes of $C_i$ and compute corresponding FASTPs.
11:       Plug the FASTPs into the equations of $C_i$ and use identity testing to check whether all equations are satisfied. If yes, keep the solution, otherwise, drop it.

---

**Checking the solutions:** If we only get one solution, then we are done (since we know that the system has at least one solution). If we get two solutions, then it could be that only one of them is valid. This can happen if there is another identifying cycle and only one of the two solutions of the first cycle is a solution of the second. (There always has to be one solution by assumption.) To check this, we simply plug the two computed solutions in all missing edge equations and check whether they satisfy all of them (line 11). An easy calculation shows that this reduces to the problem of checking whether a FASTP is identically zero. This can be done in polynomial time [van der Zander et al., 2022].

To get our faster algorithm, we improve on the running times of the first, second, and fourth item above.

## 4 A faster algorithm

We now present an improved realization of Algorithm 1. We will need to compute with polynomials. They will be represented by arithmetic circuits, which can also be implicitly given, for instance by iterated matrix products. All polynomials are defined over the reals, but they all have integer coefficients and can be represented exactly.

### 4.1 Faster computation of the covariances and rank detection

To compute the rank of an edge (line 1 of Algorithm 1), we first need to compute the covariances $\sigma_{i,j}$ as polynomials in the $\lambda_{u,v}$ and $\omega_{u,v}$. This can be done using (1). Gupta and Bläser [2024b] estimate this step with $O(n^3 \log n)$ operations. We here show that we can perform this step even faster with $O(n^2)$ operations using *Wright's trek rule* [Wright, 1934], see also [Drton, 2018, Thm 4.1].

**Lemma 1.** *Given a tree-shaped SCM $M$, we can construct an algebraic circuit of size $O(n^2)$ that computes all covariances $\sigma_{i,j}$.*

See Appendix A for the proof. Above, we produce an arithmetic circuit for the polynomials $\sigma_{i,j}$. To check whether the bidirected edge $i \leftrightarrow j$ has rank two, we need to check whether the polynomial $\det \begin{pmatrix} \sigma_{i,q} & -\sigma_{i,j} \\ \sigma_{p,q} & -\sigma_{p,j} \end{pmatrix}$ vanishes (as a polynomial). Since the entries of the $2 \times 2$-matrix have small circuits, so has the determinant. Checking whether the determinant vanishes as a polynomial is an instance of the so-called *polynomial identity testing problem (PIT)*, which can be solved efficiently in polynomial time by a randomized algorithm, the Schwartz–Zippel algorithm, see the next section.

**Remark 1.** *Since the $\sigma_{i,j}$ have only polynomially many monomials, we could explicitly expand the determinant as a sum of monomials. However, using the Schwartz–Zippel lemma will be faster. Later, we will also encounter (small sized) circuits for polynomials with exponentially many monomials. Then, the only way to perform identity tests is via the Schwartz–Zippel lemma.*

## 4.2 More efficient polynomial identity testing

**Lemma 2** (Schwartz–Zippel Lemma, see e.g. Shpilka and Yehudayoff, 2010, Lemma 4.2). *Let $f$ be a nonzero polynomial of degree $\leq d$ in $n$ variables. If we draw a vector $x \in S^n$ uniformly at random for some finite set $S$, then $\Pr[f(x) = 0] \leq \frac{d}{|S|}$.*

Each $\sigma_{i,j}$ has degree $\leq 2n - 1$, since each monomial corresponds to a trek and a trek can have at most $2n - 1$ edges. The corresponding $2 \times 2$-determinants have degree $\leq 2(2n - 1)$. Thus if we choose $x$ uniformly at random from $S = \{0, \ldots, 4(2n-1)\}$, we have a success probability of $\geq 1/2$. (Which maybe sounds bad at a first glance, but we can boost this probability by repeating the test several times or by choosing a larger set $S$.) It is a major open problem in complexity theory, whether this algorithm can be derandomized, see [Shpilka and Yehudayoff, 2010, Ch. 4].

Next, we present a new crucial speedup of multiple identity tests. The polynomial $\sigma_{i,j}$ can appear in several determinants. Whenever we perform another identity test, we have to evaluate $\sigma_{i,j}$ at a new point again. We can avoid this recomputation by not choosing a new random point every time, but choosing one point for *all* identity tests at the (small) cost of choosing the random point from a somewhat larger set.

**Lemma 3** (Generalized Schwartz–Zippel). *Let $f_1, \ldots, f_\ell$ be nonzero polynomials of degree $\leq d_\lambda$, $1 \leq \lambda \leq \ell$, and in $n$ variables each. If we draw a vector $x \in S^n$ uniformly at random from some finite set $S$, then $\Pr[f_\lambda(x) = 0$ for at least one $1 \leq \lambda \leq \ell] \leq \frac{d_1 + \cdots + d_\ell}{|S|}$.*

*Proof.* Follows directly from the Schwartz–Zippel lemma applied to the polynomial $f_1 \cdots f_\ell$. $\square$

The way to apply this generalized lemma is as follows: If we want to compute the rank of all edges, we compute the $\sigma_{i,j}$ using Lemma 1 only *once*, evaluated at a point from a large enough set according to Lemma 3. Then we use these values to compute all determinants. By Lemma 3, with high probability, each value will be nonzero iff the corresponding polynomial is nonzero. This saves a lot of arithmetic operations and brings its number down from $O(n^4)$ to $O(n^2)$.

**Remark 2.** *In the generalized Schwartz–Zippel lemma, we assume that all polynomials are independent of each other, in the sense that all the polynomials are given at once first and we then choose the random value. In the actual algorithm, the polynomials will, of course, be evaluated sequentially. However, it turns out that if all randomized identity tests that we perform yield the correct answer (which they will do with high probability), then in our algorithm, we always construct the same polynomials during the course of the algorithm. Only if a test fails, we might construct different polynomials. But then the result is incorrect anyway. Therefore, we can assume that the required independence holds in our algorithm.*

While the generalized Schwartz–Zippel lemma allows us to compute the rank of all missing bidirected edges by evaluating the arithmetic circuit from Lemma 1 of size $O(n^2)$ only once, the numbers occurring during the computation can be quite large. The $\ell_1$-norm $|p|_1$ of a polynomial $p$ is the sum of the absolute values of its coefficient. Since there are at most $\binom{n}{2}$ treks contributing to $\sigma_{i,j}$, $|\sigma_{i,j}|_1 \leq n^2$. By the submultiplicativity and subadditivity of the $\ell_1$-norm, $|\det \begin{pmatrix} \sigma_{i,q} & -\sigma_{i,j} \\ \sigma_{p,q} & -\sigma_{p,j} \end{pmatrix}|_1 \leq 2n^4$.

**Proposition 1.** *Let $p$ be a polynomial in $n$ variables of degree $d$. Let $x$ be a vector of length $n$ such that the absolute value of each entry is bounded by $s$. Then $|p(x)| \leq |p|_1 s^d$.*

The proof is in Appendix A. As a consequence, the resulting numbers in the rank computation of one matrix above might be as large as $2n^4 \cdot s^{2(2n-1)}$. This number has polynomially many bits! While we can perform computations with it in time polynomial in $n$, they do not fit into the register of a Word-RAM (for the theoretical analysis) nor in a built-in integer (in the actual implementation). Word-RAM is the standard model that is nowadays used in many standard books on algorithms theory, see, for instance, [Mehlhorn and Sanders, 2008]. Word-RAM registers can only store numbers with $O(\log n)$ bits in each register, which can be accessed in unit time. If we want to store longer numbers, we have to use many registers and this will result in a higher theoretical running time, since accessing a long number will mean accessing many registers. But it would also yield a higher practical running time, since we would need to use big-integer packages. There is a solution called algebraic fingerprinting [Schönhage, 1979]:

**Proposition 2.** *Let $z$ be a nonzero integer. If we choose a prime $p$ uniformly at random from a set of $N$ primes, then $\Pr[p \text{ divides } z] \leq \lceil \log_2 z \rceil / N$.*

Again, see Appendix A for the proof. As a consequence, instead of computing with the full numbers, it is enough to compute modulo a random prime to preserve nonzeroness. Note that since evaluation modulo a prime is a ring homomorphism, all intermediate computations when evaluating the arithmetic circuit for the determinants will be performed modulo the chosen prime and therefore, all intermediate results stay small.

In the example of the determinants above, the total degree of the product of all of them is $2(2n-1)n^2$ and the $\ell_1$-norm of the product is $\leq (2n^4)^{n^2}$. (We have to take the product of all polynomials, since we use the generalized Schwartz–Zippel Lemma.) Thus, the number $z$ will be $\leq (2n)^{4n^2} \cdot s^{2(2n-1)n^2}$ and we have $\log z \leq 4n^2 \log n + (2n-1)n^2 \log s$. Since the quantity $s$ can be chosen as polynomially bounded in $n$ by the generalized Schwartz–Zippel lemma, if we take primes of a set of, say, size $n^4$, then the probability of error goes to 0 as $n$ increases. By the prime number theorem, the primes that we need can be bounded by $O(n^4 \log n)$, i.e., they have $O(\log n)$ bits. Therefore, they fit into a register of a Word-RAM and we can count all arithmetic operations at unit costs.

Note that the calculations above are done to explain the overall reasoning. There will be further identity tests in the algorithm and we have to add all the degrees occurring there and multiply all the $\ell_1$ norms. The following lemma (proven in Appendix A) shows that for the theoretical analysis of the algorithm on a Word-RAM, it is enough that the degrees are polynomially bounded and the $\ell_1$-norms are exponentially bounded. Appendix C contains a detailed account of all identity tests performed by the actual implementation and concrete estimates of the $\ell_1$-norms and the resulting error probabilities.

**Lemma 4.** *Assume we want to perform a polynomial number of identity tests of polynomials with polynomially bounded degree and $\ell_1$-norm bounded by $2^{\mathrm{poly}(n)}$. Then there is a polynomial $s$ such that if we choose $x$ to be a vector with entries from a set of size $s(n)$ and a random prime among a set of $s(n)$ primes, then all identity tests will be correct with probability tending to 1 as $n$ goes to infinity.*

### 4.3  Further improved steps

Figure 3 shows how we improve the running times of the components of Algorithm 1 by using improved PIT. The improvement of the rank-tests (used in lines 1, 2) have already been described in Sections 4.1.

|  | Gupta & Bläser | This work | Location |
|---|---|---|---|
| Rank-tests | $O(n^3 \log n)$ | $O(n^2)$ | Sec. 4.1 |
| Cycle-detection | $O(n^4)$ | $O(n^3 \log^2 n)$ | App. B.1 |
| Self-reduction | $O(n^6)$ | $O(n^3)$ | App. B.2 |
| FASTP-testing | $O(n^6)$ | $O(n^2)$ | App. B.3 |

Figure 3: Our improved running times

Next, we have to find an identifying cycle in line 7. This is done in two steps: First, finding one node in such a cycle and its length and then finding the cycle itself using a technique called self-reduction. Let $M_{\mathrm{miss}} = (V, \bar{B})$ denote the graph of missing bidirected edges. To each missing bidirected edge correspond two Möbius transformations as in (5). We replace each missing edge $i \leftrightarrow j$ by two directed ones and give them the weights $M_{i,j} = \begin{pmatrix} \sigma_{i,q} & -\sigma_{i,j} \\ \sigma_{p,q} & -\sigma_{p,j} \end{pmatrix}$ and $M_{j,i} = -\operatorname{adj}(M_{i,j})$, see (5). The strongly connected components $\vec{C}$ formed by rank-two edges of the resulting graph are considered separately. Gupta and Bläser [2024a, Lemma 8] prove that a cycle of missing edges gives a nontrivial equation, which allows to identify all parameters on this cycle, if the product of the matrices along this cycle, which we call the weight of the cycle, is not a multiple of the $2 \times 2$-identity matrix. Such a cycle is called *identifying*. Let $W$ be the adjacency matrix of $\vec{C}$. The entries of $W$ are $2 \times 2$-matrices with polynomials as entries and testing whether a weight is a multiple of the identity matrix is an instance of PIT. The entry in position $(i, i)$ of $W^t$ is the sum of the products of the edge weights of all closed walks of length $t$ starting in node $i$. If there is an identifying cycle starting in $i$ of length $t$, then its weight is not a multiple of the $2 \times 2$-identity matrix and it is added to the entry in position $(i, i)$. However, this contribution might be canceled by other walks. Therefore, each edge $e$ also gets a label $x_e$, which is a variable, and we give $e$ the new weight $x_e \cdot M_e$. In this way, each simple walk gets a unique monomial as label and the cancellations cannot happen. Now we can use PIT to find the entry $i$ and the shortest length of an identifying cycle. This gives an $O(n^4)$ algorithm, since we have to compute all powers $W^2, \ldots, W^n$. To speed up the computation, we add a self-loop to every node.

Then the power of the adjacency matrix contains the sums of weights of all walks of length *at most* t. Then we can do binary search to find the smallest $t$ in time $O(n^3 \log^2 n)$, see Appendix B.1. The use of the generalized Schwartz–Zippel lemma is essential, since we can reuse precomputed powers of $W$ and do not recompute them with new random values.

Once we know a starting node $i_0$ of an identifying cycle and its length $t$, we can find it using a technique called self-reduction (still line 7 of Algorithm 1). We switch to a layered version of the graph, where layers correspond to time steps. Let $W_1, \ldots, W_t$ be the adjacency matrices between the layers, each is a copy of $W$. We precompute the suffixes $S_j = W_j \cdots W_t \cdot e_{i_0}$ for $1 \leq j \leq t$, where $e_{i_0}$ is the $i_0$th unit vector. This can be done in time $O(n^2 t)$. Now start removing the edges leaving $i_0$ in the first layer one after another. This means that we simply set the corresponding entry in $W_1$ to 0. Let $W_1'$ be the resulting matrix. We check whether $e_{i_0}^T \cdot W_1' W_2 \cdots W_t \cdot e_{i_0}$ is not a multiple of the $2 \times 2$ identity matrix. When we found the first edge leaving $i_0$, say it goes to node $j$, such that this is not true, then we know that $i \leftrightarrow j$ is part of an identifying cycle. We now know that $e_j^T \cdot W_2 \cdots W_t \cdot e_{i_0}$ is not a multiple of the identity and we can go on recursively. The precomputation time is $O(n^3)$. Every edge of the graph $G$ is considered at most once and each test of the edge can be done in time $O(n)$. So the overall running time is $O(n^3)$ since there are at most $n^2$ edges, see Appendix B.2 for details. The use of precomputed values again works by the generalized Schwartz–Zippel lemma.

When we found an identifying cycle, we get either a rational expression or a FASTP $\frac{p + q\sqrt{s}}{r + t\sqrt{s}}$ for one of the parameters. This can be transferred to all other parameters in $\vec{C}$ using the Möbius transformations on the edges. In the second case, we can either have one or two solutions (since there are potentially two roots). To decide this, we have to test whether the potential two solutions satisfy all other equations of $\vec{C}$. This is an instance of identity testing for FASTPs. Van der Zander et al. [2022] give such an algorithm, which is involved, since it requires testing whether a polynomial given as a circuit is a perfect square. We show that this can be avoided and we can work with ordinary PIT, improving the running time to $O(n^2)$. Altogether, we obtain the following theorem, see Appendix B.3.

**Theorem 1.** *There is a randomized algorithm that decides generic identifiability in a tree-shaped SCM in time $O(n^3 \log^2 n)$ (on a Word-RAM). For each parameter $\lambda_{p,i}$, it correctly decides whether the parameter is 1-identifiable, 2-identifiable, or not identifiable at all. (No other cases can occur.)*

## 5 Benchmarking

We provide our C++ implementation as the open-source R package *fasttreeid*, available on GitHub [Briefs and Bläser, 2025b] and CRAN [R Core Team, 2025, Briefs and Bläser, 2025a]. We tested it against the treeID-algorithm, which is part of the DAGitty package [Textor et al., 2016]. We used the current version 3.1. The treeID-algorithm is specifically made for tree-shaped SCMs and is based on [van der Zander et al., 2022]. It is the only implementation we are aware of that is specifically designed for tree-shaped SCMs. Second we also benchmarked against the half-trek criterion (HTC) [Foygel et al., 2012], which is part of the state-of-the-art R-package SEMID [Barber et al., 2023], we used the current version 0.4.1. Note that HTC is designed for arbitrary SCMs. On the other hand, it is not complete, that is, it can fail to identify parameters that are in principle identifiable. We chose HTC from SEMID, since it is the fastest identification routine of the package. All experiments were carried out on a Debian Linux server equipped with an AMD EPYC 7702 64-core processor running at 3.35GHz and an overall memory of 2TB.

### 5.1 879 graphs benchmark

Van der Zander et al. [2022] provide 879 test cases with 8 nodes each. The directed edges form a line $0 \rightarrow 1 \rightarrow 2 \rightarrow 3 \rightarrow 4 \rightarrow 5 \rightarrow 6 \rightarrow 7$. The supplementary material of [van der Zander et al., 2022] contains the full classification of all edges, therefore, we used it to check the correctness of our implementation. The result is given in the table in Figure 4.

Our implementation is faster by a factor of 5000. When running all 879 test cases within C++ directly, our program only takes 0.1 seconds, so there seems to be some overhead

| Program | Cases solved | Total execution time [$s$] |
|---|---|---|
| Our Program | 879 | 7.955 |
| treeID | 879 | 40435.533 |

Figure 4: Running times on the 879 graphs benchmark

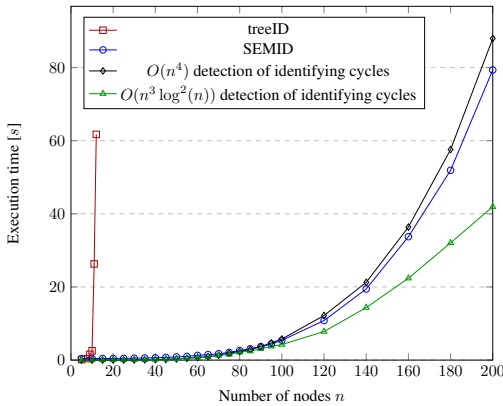

Figure 5: Random tree structure, one large cycle of missing bidirected edges

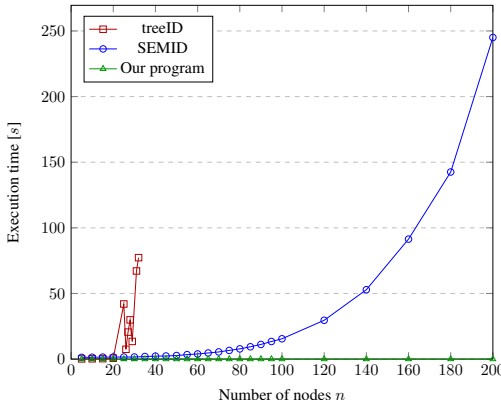

Figure 6: Tree is a path, random set of $80\%$ of the possible bidirected edges (all parameters identifiable by missing bidirected edges to 0)

from IO when invoking the program using a shell script. We suspect that the fact that our algorithm is much faster than treeID does not only come from the fact that we have a faster way to find identifying cycles: Since there are only few potential missing cycles in the 879 graphs, because the graphs are small, we suspect that it also stems from the fact that the identity tests might be implemented inefficiently in treeID (Appendix B.4).

## 5.2 One large missing cycle

The overall running time of our algorithm does not depend too much on the actual graph structure, since we mainly use linear algebra operations. The minimum length of an identifying cycle has some influence on the binary search process, while the edge density of the missing edge graph affects the self reduction step.

The first set of test cases was obtained by creating a random directed tree (each node takes as its parent a smaller node uniformly at random, this creates trees of logarithmic depth with high probability). The only missing bidirected edges form a cycle $1 \leftrightarrow 2 \leftrightarrow \cdots \leftrightarrow (n-1) \leftrightarrow 1$. All three programs were run on a single core with a time limit of 5 minutes per test case. treeID was run with 16 GiB of memory and did not terminate within the time limit for $n \geq 13$. HTC/SEMID did not solve any of the test cases, but terminated on all of them. One can formally prove that HTC cannot identify any nodes if there are no missing edges to the root, see Appendix E. Our program solved all test cases. Figure 5 shows the execution times of the respective programs. The black line shows the performance of our program with cycle detection in $O(n^4)$, the green line shows the performance with cycle detection in $O(n^3 \log^2(n))$.

## 5.3 Random graphs with different densities

Components that are connected to 0 by missing edges are easy for our program and for treeID. For our program, the execution time on such test cases is very small and dominated by generating the treks to compute the $\sigma_{i,j}$ $(O(n^2))$. Interestingly, treeID does not perform too well, we suspect that this is due to the implementation of the identity tests, see Appendix B.4. When creating a random graph with a given density, the probability that there are missing edges to 0 is high, so we consider random graphs with no missing edges to 0 separately (see Appendix D). In Figure 6, the directed edges form a path $0 \to 1 \to \cdots \to (n-1)$ and the bidirected edges are a random subset containing $80\%$ of all possible bidirected edges. In all test cases, all identifiable nodes are (1-)identifiable due to missing bidirected edges to 0, and moreover, only a few nodes are not identifiable at all. For our program, a much shorter execution time can be observed compared to Figure 5. HTC solved all test cases, but took about three times as long as in Figure 5. treeID exceeds the memory limit of 256 GiB for $n = 30$ and $n \geq 33$ (but works for $n = 31$ and $n = 32$). We have no idea why treeID takes so much longer for $n = 25$ than for $n = 26$, but this behavior is reproducible. Further benchmarks can be found in Appendix D. In all of them, our implementation outperforms both treeID and HTC.

## Acknowledgments

The authors thank Batuhan Koyuncu and Isabel Valera for helpful discussions.

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

# A Omitted proofs

**Lemma 1.** *Given a tree-shaped SCM $M$, we can construct an algebraic circuit of size $O(n^2)$ that computes all covariances $\sigma_{i,j}$.*

*Proof.* We have $\sigma_{0,0} = \omega_{0,0}$. For $\sigma_{i,0}$ with $i > 0$, we have $\sigma_{i,0} = \sigma_{p,0}\lambda_{p,i} + \omega_{i,0}$, where $p$ is the parent of $i$. This follows from (4), where in general the left-hand side is $\omega_{i,0}$. For $i, j$ both greater than 0, let $p$ and $q$ be the parents of $i$ and $j$, respectively. We have $\sigma_{i,j} = \sigma_{p,j}\lambda_{p,i} + \sigma_{i,q}\lambda_{q,j} - \sigma_{p,q}\lambda_{p,i}\lambda_{q,j} + \omega_{i,j}$. The first term contains all treks that end in $p$ and $j$ and extends them by the directed edge $p \to i$. The second term does the same for all treks that end in $i$ and $q$, respectively. We are counting all treks twice in this way that ended in $p$ and $q$ and were extended by the directed edges $p \to i$ and $q \to j$. They are subtracted once through the third term. The fourth term is the new trek which corresponds to a bidirected edge between $i$ and $j$. Alternatively, we can also use (3). (In all these computations, if a bidirected edge is missing, the corresponding $\omega$-variable is replaced by 0). The number of arithmetic operations per $\sigma_{i,j}$ is constant, therefore, the overall size of the circuit is $O(n^2)$. $\square$

**Proposition 1.** *Let $p$ be a polynomial in $n$ variables of degree $d$. Let $x$ be a vector of length $n$ such that the absolute value of each entry is bounded by $s$. Then $|p(x)| \leq |p|_1 s^d$.*

*Proof.* Each monomial of $p$ has total degree $d$. If we evaluate the monomial at a point with coordinates bounded by $s$, the maximum value we get is $s^d$. The monomials are multiplied with the coefficients and then added up, therefore, we get an upper bound by multiplying $s^d$ with the $\ell_1$-norm of $p$. $\square$

**Proposition 2.** *Let $z$ be a nonzero integer. If we choose a prime $p$ uniformly at random from a set of $N$ primes, then $\Pr[p \text{ divides } z] \leq \lceil \log_2 z \rceil / N$.*

*Proof.* The product of any $N$ pairwise distinct prime numbers is lower bounded by $2^N$. On the other hand, $z$ can have at most $\log z$ prime divisors. Therefore at most $\log z$ of the $N$ primes can divide $z$ and we get the statement of the lemma. $\square$

**Lemma 4.** *Assume we want to perform a polynomial number of identity tests of polynomials with polynomially bounded degree and $\ell_1$-norm bounded by $2^{\text{poly}(n)}$. Then there is a polynomial $s$ such that if we choose $x$ to be a vector with entries from a set of size $s(n)$ and a random prime among a set of $s(n)$ primes, then all identity tests will be correct with probability tending to $1$ as $n$ goes to infinity.*

*Proof.* Let $p(n)$ be the number of identity test and $d(n)$ be an upper bound for the degrees and assume that the $\ell_1$-norm of each polynomial is bounded by $2^{q(n)}$. By the generalized Schwartz–Zippel lemma, if we choose the entries of $x$ from a set of size $np(n)d(n)$, then the error probability of selecting a bad $x$ will tend to 0. The $\ell_1$-norm of the product polynomial in the generalized Schwartz–Zippel lemma will be bounded by $2^{q(n)p(n)}$. Therefore, by Proposition 1, the size of the number after plugging in $x$ will be $\leq 2^{q(n)p(n)} \cdot (np(n)d(n))^{d(n)}$. By Proposition 2, if we choose a prime at random from a set of $n \cdot (q(n)p(n) + d(n)\log(np(n)d(n)))$ many primes, the error probability when computing modulo this prime will go to 0, too. The overall error will go to zero, too, by the union bound. $\square$

# B Further details on the algorithm

## B.1 Faster detecting of identifying cycles

The main difficulty in Algorithm 1 is line 7, finding an *identifying* cycle of missing edges, that is, a cycle that gives a nontrivial equation for identifying a parameter. Let $M_{\text{miss}} = (V, \bar{B})$ denote the graph of missing bidirected edges. $\vec{M}_{\text{miss}}$ denotes the directed version of $M_{\text{miss}}$, in which each bidirected edge $i \leftrightarrow j$ is replaced by two directed edges $(i, j)$ and $(j, i)$. Furthermore, $(i, j)$ has a $2 \times 2$-matrix $M_{i,j} = \begin{pmatrix} \sigma_{i,q} & -\sigma_{i,j} \\ \sigma_{p,q} & -\sigma_{p,j} \end{pmatrix}$ as weight and $(j, i)$ has the matrix $M_{j,i}$, which is $-\operatorname{adj}(M_{i,j})$, see (5). The edges $(i, j)$ in the directed missing edge graph are classified into rank-1 and rank-2 edges, depending on whether the rank of $M_{i,j}$ if 1 or 2, respectively. By definition, the rank of $M_{j,i}$ is the same as $M_{i,j}$, so we can also classify the edges of $\vec{M}_{\text{miss}}$. Let $M_{\text{miss},2}$ and $\vec{M}_{\text{miss},2}$ denote the subgraph of $M_{\text{miss}}$ and $\vec{M}_{\text{miss}}$, respectively, that consists only of the rank 2 edges. Any weakly

connected component of $\vec{M}_{\mathrm{miss},2}$ is always strongly connected. In the following $\vec{C}$ denotes a strongly connected component of $\vec{M}_{\mathrm{miss},2}$, i.e., the directed version of a connected component $C_i$ considered in the for loop in line 3 of Algorithm 1.

Gupta and Bläser [2024a] give an efficient randomized algorithm to test whether there is an identifying cycle in the component $\vec{C}$. This test takes time $O(n^4)$ as reported in [Gupta and Bläser, 2024b]. We here improve this to $O(n^3 \log^2 n)$. Gupta and Bläser [2024a, Lemma 8] prove that a cycle is identifying, if the product of its edge weights $M_{e_1} \cdots M_{e_t}$ is not a multiple of the identity matrix, where $e_1, \ldots, e_t$ are the edges of the cycle. Note that this is again an instance of polynomial identity testing, since we need to check whether the entries in position $(2,1)$ and $(1,2)$ are zero and whether the difference of the entries in positions $(1,1)$ and $(2,2)$ are zero. Let $W$ denote the weighted adjacency matrix of $\vec{C}$, that is, the entry in position $(i,j)$ is the $2 \times 2$-matrix $M_{i,j}$. It is well known, see e.g. [Gupta and Bläser, 2024a, Fact 15], that the entry in position $(i,j)$ of the $t$th power $W^t$ is the sum of all weights of paths of length $t$ from $i$ to $j$. Here, the weight of a path is the product of the weights of its edges. However, it can happen that the contribution of several walks cancel each other, therefore, although there might be identifying cycles, which are walks from $i$ to $i$, the entry in position $(i,i)$ is a multiple of the identity matrix.

The basic idea to repair this is that each directed edge $(u,v)$ in $\vec{C}$ gets the weight $x_{u,v} M_{u,v}$ instead of its original matrix weight $M_{u,v}$. Here $x_{u,v}$ is an additional indeterminate. Let $\hat{W}$ be the new modified adjacency matrix. If $w$ is a walk from $u$ to $v$ consisting of edges $e_1, \ldots, e_t$, then the weight of this walk is $x_{e_1} \cdots x_{e_t} M_{e_1} \cdots M_{e_t}$. We call $\ell(w) = x_{e_1} \cdots x_{e_t}$ the label of $w$ and $M(w) = M_{e_1} \cdots M_{e_t}$ its matrix.

**Lemma 5.** *The entry of $\hat{W}^t$ in position $(i,j)$ equals*

$$\sum_{\substack{\text{walks } w \text{ from } i \text{ to } j \text{ of length } t}} \ell(w) \cdot M(w)$$

*Proof.* Follows from [Gupta and Bläser, 2024a, Fact 15] applied to $\hat{W}$. We give a proof for the sake of completeness, though. The proof is by induction on $t$. The case $t = 1$ is obvious. For $t > 1$, we can write $\hat{W}^t = \hat{W}^{t-1} \cdot \hat{W}$. Each walk from $i$ to $j$ of length $t$ is a walk from $i$ to some node $k$ of length $t-1$ extended by the edge $(k,j)$. Thus

$$\hat{W}^t_{i,j} = \sum_{\substack{\text{walks } w' \text{ from } i \text{ to } k \text{ of length } t-1}} \ell(w') M(w') \cdot x_{k,j} M_{k,j} = \sum_{\substack{\text{walks } w \text{ from } i \text{ to } j \text{ of length } t}} \ell(w) M(w).$$

$\square$

**Observation 1.** *If $w$ is a simple walk, then the label $x_{e_1} \cdots x_{e_t}$ is unique, since two simple walks have to differ in at least one edge.*

**Remark 3.** *If we have a simple closed walk $w$ of length $t$, then $w$ can have every of its nodes as a starting point, which is then also its end point. This will not change the label. However, depending on the chosen starting node $i$, $w$ will contribute to $W^t_{i,i}$, therefore, these labels will not mix up.*

**Remark 4.** *While the label will not change, the matrix $M(w)$ can change depending on its starting point $i$. Although the matrix $M(w)$ itself may change, it will not change whether $w$ is identifying or not. This is due to the fact that whenever $M_{e_1} \cdots M_{e_t} = \alpha \cdot 1$, i.e., the product is a multiple of the identity matrix, then the product of any cyclic permutation is $\alpha \cdot 1$, too. (Multiply the equation from the left with $M_{e_1}^{-1}$ and with $M_{e_1}$ from the right and repeat if needed.)*

However, two arbitrary walks can consist of the same set of edges and can still be different. To avoid this, Gupta and Bläser [2024a, Section 6] use a layered version of the graph. This increases the size of the underlying graph by a factor of $n$, resulting in the running time of $O(n^4)$. However, we observe that we only need that simple walks get different weight by the following lemma.

**Lemma 6** (Gupta and Bläser, 2024a, Lemma 21)**.** *A shortest identifying closed walk is always simple.*

**Corollary 1.** *Let $t$ be minimal such that there is an identifying closed walk. Then there is an $i$ such that the matrix power $\hat{W}^t$ in position $(i,i)$ is not a multiple of the identity matrix.*

*Proof.* Since $t$ is minimal, every identifying closed walk is simple. Let $w_0$ be such an identifying walk. Then $M(w_0)$ is not a multiple of the identity matrix. Let $i$ be a node on $w$. The entry $a$ of $\hat{W}^t$

in position $(i, i)$ is the sum of all $\ell(w) \cdot M(w)$ over all closed walks from $i$ to $i$ of length $t$. Since the label $\ell(w_0)$ of $w_0$ is unique among all these walks, the term $\ell(w_0)M(w_0)$ will not be canceled in $a$. Since $M(w_0)$ is not a multiple of the identity matrix, $a$ is not a multiple of the identity matrix. $\qquad\square$

We could now compute the powers $\hat{W}^t$, $1 \leq t \leq n$, and do identity testing to search for the smallest $t$ such that there is an identifying walk of length $t$. However these are $n$ matrix multiplications and it takes time $O(n^4)$ in total. This is not faster than [Gupta and Bläser, 2024a].

We improve this as follows. $\hat{W}^t$ corresponds to walks of length *exactly* $t$. We would like to have a matrix that corresponds to all walks of length *at most* $t$. We form a new graph $C_\circ$ with matrix $\hat{W}_\circ$, which we obtain from $\vec{C}$ and $\hat{W}$, respectively, by adding a self loop to each node with label 1 and matrix $\left( \begin{smallmatrix} 1 & 0 \\ 0 & 1 \end{smallmatrix} \right)$. Each walk $w$ of length $s$ in $\vec{C}$ gives rise to multiple walks in $C_\circ$ of any length $s' \geq s$, by inserting self loops at any place. Note that by construction, this neither changes the label nor the matrix of the walk.

**Lemma 7.** *The entry of $\hat{W}_\circ^t$ in position $(i, j)$ equals*

$$\sum_{\text{walks } w \text{ from } i \text{ to } j \text{ of length } s \leq t \text{ in } \vec{C}} \binom{t}{s} \cdot \ell(w) \cdot M(w).$$

*Proof.* In total there are $\binom{s'}{s}$ walks in $C_\circ$ of length $s'$ that correspond to $w$ of length $s$, since there are $\binom{s'}{s}$ ways to insert the self loops into $w$. Conversely, every walk $w'$ in $C_\circ$ of length $s'$ corresponds to one walk in $\vec{C}$ of some length $s \leq s'$ after removing all self loops from $w'$. $\qquad\square$

**Corollary 2.** *If there is an identifying cycle, then there is an $i$ such that the matrix power $\hat{W}_\circ^n$ in position $(i, i)$ is not a multiple of the identity matrix.*

**Theorem 2.** *We can find a (minimal) $t$ and a node $i$ such that there is an identifying cycle of length $t$ that contains $i$ in time $O(n^3 \log^2 n)$.*

*Proof.* We compute one matrix power $\hat{W}_\circ^n$, which can be done by repeated squaring with an arithmetic circuit of size $O(n^3 \log n)$. We now need to perform identity tests for the entries in positions $(i, i)$, $0 \leq i \leq n - 1$, of $\hat{W}_\circ^n$. As explained in Section 4.2, we can do them all at once by plugging in a large enough value. Furthermore, all computations will be done modulo the chosen prime, therefore, each single arithmetic operation in the evaluation of the circuit takes constant time. If we want to find the smallest length $t$ of any identifying cycle, we can now do so by binary search, which gives the additional $\log n$ factor. $\qquad\square$

For small $n$, the $O(n^4)$ algorithm will outperform the $O(n^3 \log^2 n)$ one. Our experiments show that the break-even point is around $n = 50$. We have implemented both routines, see Figure 5.

### B.2 Faster self reduction

Once we have found a node $i_0$ on an identifying cycle and its length $t$, we have to find this cycle. (This is still done in line 7 of Algorithm 1). This is done via a technique called self-reduction. In the algorithm by Gupta and Bläser [2024b], this takes time $O(n^4 e) = O(n^6)$, where $e$ is the number of missing edges. We here give a significantly improved procedure with running time $O(n^3)$.

Let $t$ be the length of a shortest identifying cycle and assume that the node $i_0$ is contained in a shortest identifying cycle. $t$ and $i_0$ can be found using Theorem 2. To find the identifying cycle, we will use self-reducibility. It will be beneficial to switch to the layered graph as in [Gupta and Bläser, 2024a, Section 6]: Every node $v$ is replaced by $t + 1$ copies $v_0, \ldots, v_t$. All nodes with the same index $\tau$ form a layer. Edges only go from one layer to the next, that is, if there is an edge $(v, u)$ in $\vec{C}$, we will have edges $(v_\tau, u_{\tau+1})$ for $0 \leq \tau < t$ in the layered graph. We can think of the layers as "time steps", the edge from layer $\tau$ to $\tau + 1$ is the edge taken in the $(\tau + 1)$th step of the walk. Let $W_1, \ldots, W_t$ be the weighted adjacency matrix corresponding to the layers of the graph, an edge $(v_\tau, u_{\tau+1})$ has the weight $x_{v,u,\tau} M_{v,u}$. We know that $e_{i_0}^T \cdot W_1 \cdots W_t \cdot e_{i_0}$ is not a multiple of the identity matrix, where $e_{i_0}$ is the $i_0$th unit vector, since there is an identifying cycle containing $i_0$. Note that since the entries

---

**Algorithm 2** Faster self-reduction

---

**Input**: A node $i_0$ known to lie on a shortest identifying cycle and its length $t$
**Output**: An identifying cycle of length $t$ containing $i_0$

1: Compute the suffixes $S_j = W_j \cdots W_t \cdot e_{i_0}$ for $1 \le j \le t$.
2: Set $L = \emptyset$.
3: Set $M = \left(\begin{smallmatrix} 1 & 0 \\ 0 & 1 \end{smallmatrix}\right)$. // Product of the matrices of the edges found so far.
4: Remove the edges leaving $i_0$ one after another (using Lemma 8) until the identity test does not report that there is an identifying cycle.
5: Add the edge $(i_0, j)$ found to $L$
6: Let $M = M \cdot M_{i_0,j}$.
7: Repeat with the next layer.

---

of the matrices are $2 \times 2$-matrices, the entries of $e_{i_0}$ are also $2 \times 2$-matrices, the identity matrix in position $i_0$ and the zero matrix in all other positions.

Algorithm 2 describes our improved procedure: We precompute the suffixes $S_j = W_j \cdots W_t \cdot e_{i_0}$ for $1 \le j \le t$, where all variables are replaced with our random values and all computation are reduced with our chosen prime. This can be done in time $O(n^2 t)$, since these are only matrix-vector products. Now start removing the edges leaving $i_0$ in the first layer one after another. This means that we simply set the corresponding entry in $W_1$ to 0. Let $W_1'$ be the resulting matrix. We check whether $e_{i_0}^T \cdot W_1' W_2 \cdots W_t \cdot e_{i_0}$ is not a multiple of the identity matrix. $S_2 = W_2 \cdots W_t \cdot e_{i_0}$ is already precomputed. $e_{i_0}^T \cdot W_1'$ can be computed from $e_{i_0} W_1$ in time $O(1)$ by the lemma below. Thus we can compute $e_{i_0}^T \cdot W_1' W_2 \cdots W_t \cdot e_{i_0}$ in time $O(n)$ as $e_{i_0}^T \cdot W_1' \cdot S_2$.

**Lemma 8.** *Let $u$ be a vector and $A$ be a matrix. Let $A'$ be the matrix obtained from $A$ by setting the entry in position $(i, j)$ to 0. From $v = u^T A$, we can compute $v' = u^T A'$ in time $O(1)$.*

*Proof.* We can write $v_j' = v_j - u_i a_{i,j}$ and $v_k' = v_k$ for $k \ne j$. $\qquad\square$

When we found the first edge leaving $i_0$, say it goes to node $j$, then we know that $i_0 \leftrightarrow j$ is part of a missing cycle. Now we know that $M_{i_0,j} e_j^T \cdot W_2 \cdots W_t \cdot e_{i_0}$ is not a multiple of the identity and we can go on recursively.

The precomputation time is $O(n^3)$. Every edge of the graph $\vec{C}$ is considered at most once and each test of the edge can be done in time $O(n)$. So the overall running time is $O(n^3)$ since there are at most $n^2$ edges. Thus we have proved the following theorem.

**Theorem 3.** *Given a node $i_0$ and an integer $t$ such that a $i_0$ lies on an identifying cycle of length $t$, then Algorithm 2 returns the edges of an identifying cycle of length $t$ containing $i_0$ in time $O(n^3)$.*

### B.3 Simplified identity testing of FASTPs

Once we have found an identifying cycle, we get a solution for some $\lambda_{p,i}$, given as a rational expression or a FASTP [Gupta and Bläser, 2024a, Lemma 8]. We propagate this solution to all other parameters in the connected component $C$ using the Möbius transformation along the edges (lines 9 and 10 of Algorithm 1). This takes time $O(n)$. If there are two solutions for the parameter, we have to check whether both of them satisfy all equations (line 11). Gupta and Bläser [2024b] report a time complexity of $O(n^4 \cdot e) = O(n^6)$ for this step. We here improve this running time to $O(n^2)$. This improvement comes first from the fact that we use our improved identity testing scheme that only chooses one random point for all tests and second that we can avoid a square root computation in the identity tests for FASTPs.

Appendix B.4 explains the details how to perform identity testing for FASTPs in general. We can avoid the square root computation by the following observation. Since the observed covariances are generated by the underlying model, we know that there is always at least one solution. For a missing edge, we are given two candidate solutions as FASTPs, which have the same square root term $\sqrt{s}$ by [Gupta and Bläser, 2024a, Remark 5]. When we plug these into the missing edge equation (3), we get again a FASTP with square root term $\sqrt{s}$. We now need to check whether a term of the form $p \pm \sqrt{s}q$ is zero for both possibilities of the sign. If $s = 0$, then there is only one square root. So we can assume that $s$ is not identically zero. Since we know that there is one solution, $p \pm \sqrt{s}q = 0$ can

only have two solutions if both $p$ and $q$ are identically zero. Otherwise, there is only one solution. So in total, we just have $O(n^2)$ many tests, each one requiring constant time.

**Lemma 9.** *Testing whether a component $\vec{C}$ has one or two solutions (line 10 and 11 of Algorithm 1) can be done in time $O(n^2)$.*

*Proof of Theorem 1.* The correctness of our algorithm follows from the correctness proof by Gupta and Bläser [2024a], since we follow the blueprint of the algorithm and present faster implementations of each step. The running time of our algorithm follows from Table 3. □

### B.4 Details on identity testing of FASTPs

We give some further details how general testing of FASTPs works for the sake of completeness. Consider a FASTP $\frac{p+q\sqrt{s}}{r+t\sqrt{s}}$. It is sufficient to perform the identity test of the denominator and numerator separately, since we first need to check whether the denominator is nonzero and the numerator is zero. Therefore consider a term of the form $p + q\sqrt{s}$. If $s$ is not a square, then $p + q\sqrt{s} = 0$ iff $p = 0$ and $q = 0$. If $s$ it a perfect square, that is, $s = u^2$ for some polynomial $u$, then $p + q\sqrt{s} = 0$ iff $p \pm qu = 0$ (depending on which of the two possible roots $\pm u$ we have chosen for $\sqrt{s}$). The identity test for FASTPs proposed by van der Zander et al. [2022] therefore decides whether $s$ is a perfect square. If $s$ is not a perfect square, then it checks whether $p = 0$ and $q = 0$. If $s$ is a perfect square, then it computes a root $u$ (as a circuit) and checks whether $p \pm qu = 0$.

Note that the TreeID algorithm as currently implemented in the DAGitty package also avoided this square root computation. Van der Zander et al. [2022, Section B] write: "Rather than using PIT on FASTPs, we have fully expanded the equations in the CAS." For small $n$ this does not make too much of a difference, however, even for moderate $n$, the expanded polynomials become really huge. This might explain why the TreeID implementation does not perform too well once the graph size is about 20, even when the given graph has only few missing cycles, which should be a good situation for TreeID, since it enumerates all potential missing cycles.

## C Estimates

We give error estimates for the values $n = 200$, when the observed running time is about $40s$ in the worst case (see Figure 5) as well as $n = 1000$. In the latter case, the running time is about $9500s$, i.e., more than two hours.

To lower the error probability, one can run the algorithm several times with new random numbers. The error bound will drop exponentially in the number of runs. Alternatively, one can enlarge the size of the random primes used. Again, with each new bit, the error rate decreases exponentially. However, at the moment, this requires using a big integer package, since we are at the limit of built-in integers.

Using built-in integer computations instead of symbolic polynomial manipulations is the key to the speed of our algorithm.

### C.1 Estimation of the number of identity tests and the corresponding degrees

We assume that the number of bidirected edges and the number of missing bidirected edges is $n^2$ for simplicity, since this is the worst-case for all estimates. All estimates are upper bounds.

**Rank computation:** For each missing edge we perform 4 checks of degree $2n - 1$ to check if the rank is 0 and one check of degree $2(2n - 1) = 4n - 2$ to check if the rank is 1. This means that the rank computation contributes $5n^2$ identity tests, $4n^2$ of degree $2n - 1$ and $n^2$ of degree $4n - 2$.

**Cycle detection:** The worst case for the number and degree of the checks is one component of size $n$. In this case, for each cycle length and node, we have 2 checks of degree $n(2n-1)+n = 2n^2$ (product of $n$ matrices of degree $2n - 1$ and $n$ indeterminates) each to check whether the corresponding entry is a multiple of the identity matrix, so cycle detection amounts to $2n^2$ checks of degree $2n^2$ each.

**Self reducibility:** Just like cycle detection, self reducibility amounts to $2n^2$ checks of degree $2n^2$ each.

**Checking the solutions:** For each node, we check whether $a = 0$ [Gupta and Bläser, 2024a, Lemma 8, case 2]. In the component, FASTPs are propagated to all nodes via shortest paths. Each propagation adds $2n - 1$ (degree of the $\sigma_{i,j}$) to the degree of $p, q, r, t$ in the FASTP, so their total degree is $2n^2 + n(2n - 1) = 4n^2 - n$. This means $n$ identity tests of degree $4n^2 - n$ each. Once per component, we check whether the discriminant is $0$ [Gupta and Bläser, 2024a, Lemma 8, case 1]. This means at most $n$ identity tests of degree $4n^2$. Once per missing edge, we check whether both options satisfy the corresponding equation. This requires two identity tests (getA, getB). $A$ is of degree $2(4n^2 - n) + 4n^2 + 2n - 1 = 12n^2 - 1$ and $B$ is of degree $2(4n^2 - n) + 2n - 1 = 8n^2 - 1$. This means $n^2$ identity tests of degree $12n^2 - 1$ and $n^2$ identity tests of degree $8n^2 - 1$.

Summing up these values, we get an upper bound of $5n^2 + 2n^2 + 2n^2 + n + n + n^2 + n^2 = 11n^2 + 2n$ identity tests. The maximal degree is $12n^2 - 1$ and the sum of all degrees is

$$4n^2 \cdot (2n - 1) + n^2 \cdot (4n - 2) + 2n^2 \cdot 2n^2 + 2n^2 \cdot 2n^2$$
$$+ n \cdot (4n^2 - n) + n \cdot 4n^2 + n^2 \cdot (12n^2 - 1) + n^2 \cdot (8n^2 - 1)$$
$$= 8n^3 - 4n^2 + 4n^3 - 2n^2 + 4n^4 + 4n^4$$
$$+ 4n^3 - n^2 + 4n^3 + 12n^4 - n^2 + 8n^4 - n^2$$
$$= 28n^4 + 20n^3 - 9n^2$$

For $n = 1000$, for example, the value of this polynomial is $\leq 3 \cdot 10^{13}$. For $n = 200$, it is $\leq 5 \cdot 10^{10}$.

### C.2 Estimation of the $\ell_1$ norm

**Rank computation:** Each trek contributes one monomial with coefficient $1$ to one of the $\sigma_{i,j}$. For a fixed $\sigma_{i,j}$, there are at most $n \cdot (n - 1)/2 + n \leq n^2$ treks, so the $\ell_1$ norm of the $\sigma_{i,j}$ is at most $n^2$.

During rank computation, we check whether the $\sigma_{i,j}$ are nonzero and whether the determinants of the $2 \times 2$ matrices corresponding to the missing bidirected edges are nonzero. Checking whether the $\sigma_{i,j}$ are nonzero amounts to a total $\ell_1$ norm of at most $(n^2)^{n^2}$. Checking whether the determinants are nonzero amounts to a total $\ell_1$ norm of at most $(2n^4)^{n^2}$.

**Detection and self-reduction:** The entries of $W^t$ have an $\ell_1$ norm of at most $2^n n^n (n^2)^n = (2n^3)^n \leq (2n)^{3n}$ (multiplying by the indeterminates does not change the $\ell_1$ norm). During cycle detection, there are up to $n^2$ checks whether $a \neq 0$ (nodes $\times$ cycle length), which contributes at most $((2n)^{3n})^{n^2}$ to the total $\ell_1$ norm. Additionally, there are up to $n^2$ checks whether $b \neq c$, which contributes at most $(2(2n)^{3n})^{n^2}$ to the total $\ell_1$ norm. During self reducibility, there are the same kind of checks.

**Checking the solutions:** Once per component, we check whether the discriminant is $0$. The total $\ell_1$ norm of this is at most $(2(2n)^{3n} + 4(2n)^{3n})^{2n} = (6(2n)^{3n})^{2n}$. The $\ell_1$ norm of $p, q, r, t$ of the first FASTP (the node for which the cycle was found) is $2(2n)^{3n}$. In each propagation, the $\ell_1$ norm of $p, q, r, t$ is multiplied by at most $2n^2$, which amounts to a total $\ell_1$ norm of at most $2(2n)^{3n}(2n)^{2n} = 2(2n)^{5n}$ for the $p, q, r, t$ in the FASTPs. For each node, we check whether $a = 0$. The total $\ell_1$ norm of this is at most $(6(2n)^{3n})^{2n}(2(2n)^{5n})^2 + (2(2n)^{5n})^2 = (6(2n)^{3n})^{2n}4(2n)^{10n} + 4(2n)^{10n}$ ($s \cdot t^2 - r^2 = 0$). For each edge, we check whether $A$ and $B$ are nonzero. The $\ell_1$ norm of each of these $A/B$ is at most $2(2n)^{5n}2(2n)^{5n}(6(2n)^{3n})^{2n}n^2 = 4(2n)^{10n}(6(2n)^{3n})^{2n}n^2$ and there are at most $2n^2$ such checks, so $(4(2n)^{10n}(6(2n)^{3n})^{2n}n^2)^{2n^2}$ in total.

Hence, the product of all polynomials for which we do identity testing has $\ell_1$ norm at most

$$(n^2)^{n^2} \cdot (2n^4)^{n^2} \cdot ((2n)^{3n})^{n^2} \cdot (2(2n)^{3n})^{2n^2} \cdot (6(2n)^{3n})^{2n} \cdot 2(2n)^{5n}$$
$$\cdot ((6(2n)^{3n})^{2n}4(2n)^{10n} + 4(2n)^{10n}) \cdot (4(2n)^{10n}(6(2n)^{3n})^{2n}n^2)^{2n^2}$$
$$\approx 2^{12n^4 + 33n^3 + 19n^2 + 19n + 3} \cdot 3^{4(n^3 + n)} \cdot n^{n(12n^3 + 29n^2 + 22n + 15)}.$$

The logarithm is bounded by

$$(12n^4 + 33n^3 + 22n^2 + 23n + 3)(1 + \log_2 n).$$

For $n = 1000$, this is $\leq 1.4 \cdot 10^{14}$. For $n = 200$, this is $\leq 1.7 \cdot 10^{11}$.

### C.3   Error estimation

We choose a random prime with 59 bits. By the prime number theorem, there are at least $1.3 \cdot 10^{16}$ such prime numbers. (Wolfram alpha provides such values.) The prime is generated using the OpenSSL package (version 3.0.13, Jan 2024) [OpenSSL Project, Jan 2024]. Alternatively, one can also use GMP, the GMP multiprecision arithmetic library [Granlund and the GMP development team, 2023].

**Choosing a non-prime:** This can happen with probability $2^{-128}$ according to the openssl specification.

**Choosing a bad point:** The total degree of all polynomials is bounded by $3 \cdot 10^{13}$ ($n = 1000$) and $5 \cdot 10^{10}$ ($n = 200$). We pick random values from a set of size $2^{58} \approx 3 \cdot 10^{17}$ in the generalized Schwartz–Zippel Lemma. This size is limited by the size of built-in integers. Thus the error here is bounded by $10^{-4}$ (for $n = 1000$) and $1.6 \cdot 10^{-7}$ ($n = 200$).

**Choosing a bad prime:** We have to estimate the probability that a random prime divides the product of the evaluations of all polynomials at the chosen random point. For this, we have to estimate $\log_2(Ls^D)$, where $L$ is the product of all $\ell_1$-norms, $D$ is the sum of all degrees and $s$ is the size of the smallest prime. This can be estimated by the preceding items by

$$1.4 \cdot 10^{14} + 3 \cdot 10^{13} \log_2(2^{58}) \leq 1.9 \cdot 10^{15} \qquad (n = 1000)$$
$$1.7 \cdot 10^{11} + 5 \cdot 10^{10} \log_2(2^{58}) \leq 3.1 \cdot 10^{12} \qquad (n = 200)$$

The error is bounded by $(1.9 \cdot 10^{15})/(1.3 \cdot 10^{16} \cdot 58) \leq 0.0026$ for $n = 1000$ and $(3.1 \cdot 10^{12}/(1.3 \cdot 10^{16} \cdot 58) \leq 4.1 \cdot 10^{-6}$ by Proposition 2. (Note that we can divide by the extra $58 = \log_2(2^{58})$, since the smallest prime has 58 bits whereas the proof of the proposition just assumes that it is 2.)

**Overall error:** By the union bound, the overall error probability is sum of all three probabilities, which is dominated by the third one. Since there is some slack in the estimates, one can safely estimate the sum by the last error probability. So for $n = 1000$, the overall error probability is $\leq 0.0026$ and for $n = 200$, the overall error probability is $\leq 4.1 \cdot 10^{-6}$.

While for $n = 200$, the error probability might be acceptable, one has to run the algorithm a few times for $n = 1000$ to reduce the error probability. But all estimates above are very conservative worst-case estimates. For instance, the proof of the Schwartz–Zippel lemma assumes that all roots of all polynomials occurring are integers. So, the typical error probability is much lower than our worst-case estimates.

## D   Further benchmarking

### D.1   Random graphs of density 0.8, random tree structure

These test cases have a random tree structure and a random set of $80\%$ of all possible bidirected edges. In all test cases, all identifiable nodes are identifiable due to missing bidirected edges to 0, and in all of them, most of the nodes are (1-)identifiable. Both SEMID and our program solve all test cases, and our program takes less than 0.5 seconds on each test case. treeID solved all test cases up to $n = 30$ in less than 5 seconds each, but for $n = 31$ already, it failed with JavaScript heap out of memory after about 3 minutes even when given 256 GiB of memory. Figure 7 shows the execution times of the programs.

### D.2   Random graphs of density 0.2, random tree structure, no missing edges to 0

It can be proven that these test cases cannot be solved by the half-trek criterion, see Appendix E. Due to the large number of missing edges, all test cases can be identified by three-cycles (in many cases

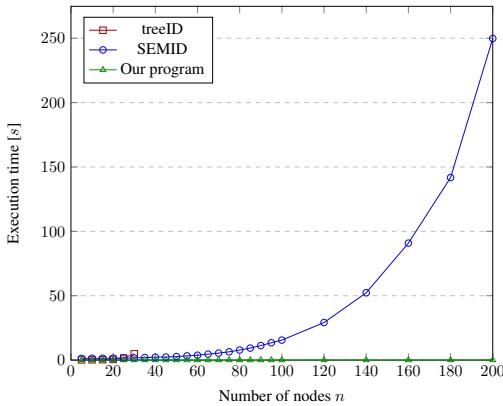

Figure 7: Random tree structure, random set of $80\%$ of the possible bidirected edges (all identifiable by missing bidirected edges to 0)

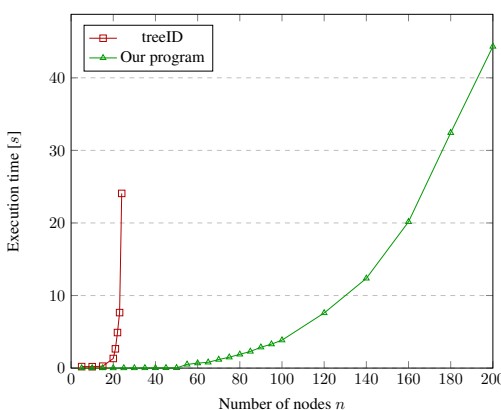

Figure 8: Random tree structure, no missing edges to 0; apart from that random set of $20\%$ of the possible bidirected edges

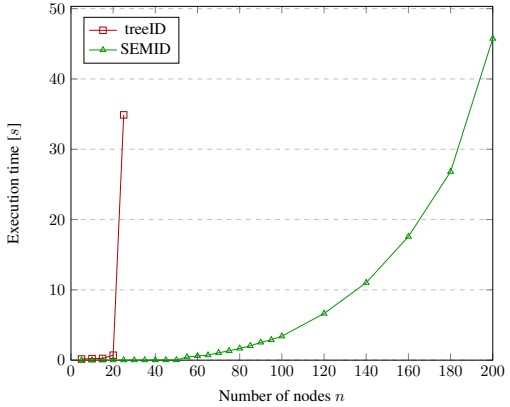

Figure 9: Random tree structure, no missing edges to 0; apart from that random set of $50\%$ of the possible bidirected edges

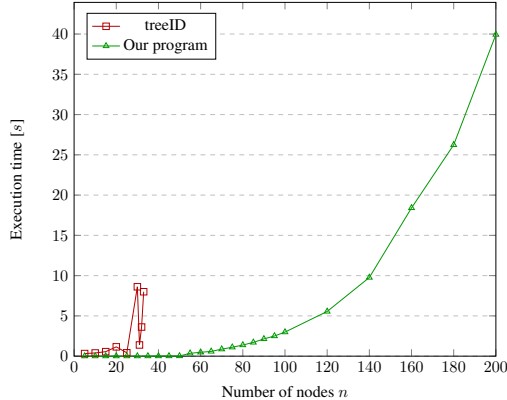

Figure 10: Random tree structure, no missing edges to 0; apart from that random set of $90\%$ of the possible bidirected edges

even $1 \leftrightarrow 2 \leftrightarrow 3 \leftrightarrow 1$). treeID runs out of memory (memory limit: 256 GiB) for $n \geq 25$. We did not benchmark against HTC/SEMID since it provably cannot identify any edges and its running times were fairly oblivious to the graph structure. The results can be found in Figure 8.

### D.3 Random graphs of density 0.5, random tree structure, no missing edges to 0

treeID seems to perform better with higher density, most likely because there are less missing edges and therefore less identifying cycles to check. treeID runs out of memory (memory limit: 256 GiB) for $n \geq 26$. The results can be found in Figure 9.

### D.4 Random graphs of density 0.9, random tree structure, no missing edges to 0

There seems to be a bug in treeID. For $n = 20$, it crashes with `if (newfastp.length == 0) throw "Inconsistent solutions"`, but when run with `--trace-uncaught` (to show where the exception was thrown), no error occurs and the output is correct. treeID runs out of memory (memory limit: 256 GiB) for $n \geq 34$. The results can be found in Figure 10.

## E   Limitations of HTC with no missing edges to the root

We explain that HTC fails to identify any parameter at all when there are no missing edges to the root. Recall that HTC works as follows: A set $Y \subset V$ satisfies the halftrek criterion with respect to $v \in V$

if $|Y|$ equals the number of parents of $v$, $Y$ does not contain $v$ nor any of the nodes that are connected by a bidirected edge to $v$, and there is a system of halftreks with no sided intersections from $Y$ to the parents of $v$. A halftrek here is a trek whose left part consists of a single node. Two treks have a sided intersection, if the left parts have a node in common or the right parts have a node in common or both.

A family $Y_v$, $v \in V$, of subsets of nodes satisfies the half-trek criterion if for every $v$, $Y_v$ satisfies the half-trek criterion with respect to $v$, and in addition, there is a total ordering of the nodes such that $w < v$ whenever $w \in Y_v \cap \mathrm{htr}(v)$ where $\mathrm{htr}(v)$ are all nodes $u$ that can be reached by a half-trek from $v$ and that are neither $v$ nor connected with a bidirected edge to $v$. Foygel et al. [2012, Thm 1] prove that if an SCM satisfies the halftrek criterion, then all parameters are rationally identifiable.

**Proposition 3.** *In a tree-shaped SCM with no missing edges to the root, HTC fails to identify any parameter.*

*Proof.* Since there are no missing bidirected edges to $0$, all nodes can reach each other via halftreks. In particular, every node $w$ that is a candidate for $Y_v$ is also contained in $\mathrm{htr}(v)$. Now assume that there is a total ordering on the nodes. For the minimum node $v_0 \neq 0$ in this ordering, there is no set $Y_{v_0}$, since there is no smaller node that could be put into $Y_{v_0}$. □

This is a drawback of all state-of-the-art identification methods on general graphs. Van der Zander et al. [2022, Proposition 2] show that any edge that is identifiable by the ACID algorithm [Kumor et al., 2020] on tree-shaped SCMs is identifiable via a missing edge to the root and then potential propagation via a Möbius transform along directed edges. Note that ACID subsumes all known polynomial time methods for generic identification in arbitrary SCMs.

