# OpenReview forum: "Faster Generic Identification in Tree-Shaped Structural Causal Models"
_NeurIPS.cc/2025/Conference — NeurIPS 2025 poster_

### Official Review · Reviewer_hSv9 · 2025-06-17

**Clarity:** 2
**Significance:** 2
**Originality:** 3
**Rating:** 5
**Confidence:** 3

**Summary:**

This paper presents a randomized algorithm with an improved time complexity from $O(n^{6})$ to $O(n^{3}(log(n))^{2})$ that is complete for identifying the functional parameters in linear structural causal models in which the directed edges form a tree. The key contribution of this paper is to develop a more efficient method for polynomial identity testing (lines 238-242) via Lemma 3. It also provides a theoretical analysis of the running time of the proposed method via algebraic fingerprinting.

**Questions:**

- Why did the authors not compare their algorithm with that by Gupta and Blaser [2024a] instead of [van der Zander et al. 2022] and [Barber et al., 2023]?
- Line 120: ‘Our implementation vastly outperforms the treeID algorithm of the DAGitty package with respect to running time.’ Does that mean the difference is only due to implementation, but not the algorithmic improvement?
- In the proof of Lemma 1, isn’t $\omega_{0,0}$ ill-defined in a mixed graph?
- In line 334, since the layers correspond to time steps, I am a bit perplexed by the use of the index t for $W_t$, is that the same as the length t?
- How can we know the difference of the execution time in Figure 4 is not affected by the choice of programming languages, e.g., C++ vs R that are used to implement the algorithms?
- How many test cases were used per number of nodes in Figure 5, similarly in Figure 6? I am guessing the y-axis execution time is the sum of all execution times of all cases? If not, shouldn’t there be variations?

Suggestions:

- Typos: Line 121: ‘eges’ -> ‘edges’
- I find it very confusing to see the index 0 represent the root in the SCM and the actual number 0 appear in equations 3 and 4 at the same time.
- I think the proofs can be more thorough without skipping steps. For example, in the proof of Lemma 1, the authors should explain why $\sigma_{i,0} = \sigma_{p,0}\lambda_{p,i} + \omega_{i,0}$? I think it should be something like this $Cov(X_{i},X_{0}) = \sigma_{i,0} = Cov(\lambda_{p,i}X_{p}+\epsilon_{i}, X_{0}) = Cov(\lambda_{p,i}X_{p} + \epsilon_{i}, X_{0}) = \lambda_{p,i}Cov(X_{p}, X_{0}) + Cov(\epsilon_{i}, X_{0})$ given the condition that X_{i} has a unique parent under a tree-shaped SCM. Also, in the proof of Proposition 2, it would be better to elaborate on why the integer can have at most log z distinct prime divisors to increase clarity.
- Line 309: ‘This is done in two steps: First finding’ -> ‘This is done in two steps: first, finding’
- In lines 323, it will be good if the authors can give an example to illustrate how the contribution may be cancelled by other closed walks

I am willing to increase my score if the authors can address my concerns well.

**Ethical Concerns:**

["NO or VERY MINOR ethics concerns only"]

**Final Justification:**

Previously, my concerns were mostly about clarity, especially for the experimental section. After the rebuttal, the authors have clarified why the baseline algorithm is not included, and I think they have provided a reasonable answer. Based on the authors' willingness to accept my suggestions and the promised revision, I believe the experimental section will now be better prepared for a wider range of audiences to understand.

**Limitations:**

yes

**Quality:**

2

**Strengths And Weaknesses:**

**Strengths:**

- The paper improves the Algorithm proposed by Gupta and Blaser in several aspects:
   - It replaces the step of computing the rank of an edge with a more efficient method using Wright’s trek rule to reduce the time complexity from $O(n^{3}\log n)$ to $O(n^{2})$.
   - It uses the Schwartz-Zippel lemma to achieve faster polynomial identity testing. The idea of performing multiple identity tests is innovative. Lemma 3 is the result that follows the idea. It brings down the complexity from $O(n^{4})$ to $O(n^{2})$.
  - The authors clearly explain how to possibly handle the large number of rank computations with algebraic fingerprinting to substantiate their algorithmic contribution.
  - For finding identifying cycles, the simple trick of adding a label on the edge to avoid cancellation is creative.
  - The authors show that ordinary polynomial identity testing is more efficient for FASTPs.

- Generic identification is an important and hard problem. This work makes the existing algorithm more efficient to scale in a restrictive case.

- I have checked the proofs and I couldn’t find any glaring issues other than some relatively minor clarity issues.

**Weaknesses:**

- The algorithm is not compared with the algorithm by Gupta and Blaser empirically, while that is the main baseline. I think even if the code is not available publicly, the authors should be able to code up this algorithm, as it is a closely related work.

- I find the space in the paper can be better utilized to enhance the quality of the discussion in the paper.
  - For example, instead of describing what specific programming languages the baselines are written in. The authors could have used to space to discuss things that are more important, e.g., writing the definition of a trek or giving the short proofs in the main paper. Also, if the introduction has already said that Gupta and Blaser improved treeID, there is no need to say ‘Our implementation vastly outperforms the treeID algorithm of the DAGitty package with respect to running time.’
  - I think the following sentences can be moved to the experiments section, and the authors can keep the key takeaway message, which is that the proposed algorithm outperforms the baselines. Otherwise, I think it makes the introduction less clear.  It unnecessarily leaves readers the room to question the significance of the contributions. For example, why do the authors need to hand-pick scenarios to show that the proposed algorithm is better? Are those scenarios significant? If so, why?
       -   ‘…our algorithm outperforms HTC, too. On certain worst case instances for our algorithm, we are still faster than HTC. However, if one wants a smaller error probability than 0.000005, we have to run our algorithm a few times and we are somewhat slower than HTC. On the other hand, HTC fails to identify any nodes on these graphs, so the results are incomparable.’
  - The second section ‘Generic identification’ repeats many definitions that have already been discussed in the introduction e.g., directed edge, bidirected edge, linear SCM, etc.
  - Section 3 defines what has been defined in the introduction as well,l e.g., the meaning of completeness.
  -  Line 176 “The conference version does not give any concrete bound, the arXiv version [Gupta and Bläser, 2024b] states a running time of $O(n^6)$.” I am not sure why there is a need to elaborate on this point in the main paper.
  -  The space can also be used to put on some definitions for improving readability. For example, some wordings are not defined in the introduction, and they appear later in section 2, e.g., in line 121: ‘…Missing bidirected edges to the root…’, it is not clear whether the authors simply mean there is no bidirected edge adjacent to the root until I have read section 2. The idea of a trek is frequently used, but its definition is missing from the paper. Other concepts like simple walk, strongly connected components can also be defined.
  - The subsection 5.1 title is very strange.

---

> ### Author Rebuttal · Authors · 2025-07-30
>
> Thank your for your thoughtful comments:
>
> Regarding weaknesses:
>
> - We are not aware of any existing implementation of the algorithm by Gupta and Bläser. With some additional effort, we could have implemented it within our code framework. However, as already pointed out in our response to Reviewer 8jdN, the algorithm will not be practical even for moderate values of $n$. We provide more details in the answer to your first question below.
>
> - Thank you for your suggestions for improving the presentation. We will take them into account and incorporate them. Since the paper lies at the intersection of algebraic statistics and causality and employs methods from algebraic algorithms and computer algebra, many concepts from different areas need to be introduced. Depending on the background of the actual reader, concepts might be more or less well-known. So we are really grateful for your input.
>
> Regarding questions (our answer items follow the same order as the question items):
>
> - We did not compare our algorithm with that of Gupta and Bläser for two main reasons. First, we are not aware of any existing implementation of their algorithm. Their result is primarily of theoretical interest, identifying a graph class for which generic identification is provably feasible in the sense of algorithm theory. Second, their algorithm, in its current form, is not practical even for moderate $n$. With some additional effort, we could have adapted our framework to implement the Gupta and Bläser algorithm. However, the running time of their algorithm is $O(n^6)$, while ours is $O(n^3 \log^2 n)$. As shown in Figure 5, for $n = 100$, our algorithm already performs clearly below $O(n^4)$ (with the break-even point around $n = 50$). Given that both algorithms use similar techniques, the constants hidden in the $O$-notation are likely comparable. Thus, our implementation is faster by a factor of at least $n^2$ on moderately sized instances — for example, a factor of $10.000$ for $n = 100$.
>
> - Both factors contribute. The treeID algorithm has exponential worst-case running time as it brute-forces over all potential identifying cycles, which is one reason why our method is faster. Additionally, the implementation of identity tests in treeID appears to be inefficient. From our understanding of the treeID code, identity tests are performed by expanding polynomials and passing them to a computer algebra system. This approach can take exponential time (and even space) in the worst case. We believe this, at least partly, explains the irregular behavior observed when $n$ reaches 20.
>
> - $\omega_{0,0}$ denotes the variance of the error term $\epsilon_0$ (see line 31) at the root node $0$. $\omega_{0,0} = \sigma_{0,0}$, since the root has no incoming directed edges. One could have used the notation $\omega_0$, but using double indices makes the summation formulas derived from Wright’s trek rule more elegant. (Similarly, $\omega_{i,i}$ denotes the variance of the error term at node $i$.) We will clarify this in the revised version.
>
> - $t$ always means the length of an identifying cycle, which we are currently looking for. This is stated in line 321. In the layered graph in the self reduction part, time steps go from $1$ to $t$, so $t$ is the "total time". We understand your confusion and maybe it is better to rename $t$ into $\ell$ for length.
>
> - Of course, running times are affected by the choice of programming language and other implementation details. However, these factors only influence the leading constants in the running time. If we have only one input size as in Figure 4, it is very hard to estimate the influence of the different effects. But as shown in Figure 5, we are also faster when $n$ increases, which means that the main effect comes from our faster algorithm. The reason we tested our algorithm on the benchmark of 879 graphs in Figure 4 was that the expected output was known from the paper by van der Zander et al. (2022), and we used it to verify correctness.
>
> - The runtime of our algorithm is very stable, and two main features influence it: whether there are missing edges to the root, and the length of the identifying cycles. Beyond that, the specific graph structure has little impact. If there are missing edges to the root, we can identify almost immediately (as shown in Figures 6 and 7). After that, the runtime depends only on the length of the missing cycle, since we use iterated matrix products to search for all potential missing cycles in parallel. However, this influence is also small, since we use binary search and we always need to check for length $n/2$. In the worst case of length $n$, we will always branch in the more costly half of the binary search, but the overall effect is minor. Similar observations apply to the runtime of the treeID algorithm, aside from the inefficiencies in identity testing, and HTC. Execution times were consistent across multiple runs. For simplicity, we used a single input instance to generate the plots, as noted under "7. Experiment statistical significance" in the paper checklist. The actual edge distribution can influence the running time somewhat, but only lower order terms: the tree structure slightly affects variance computation (an $O(n^2)$ term), and the density of bidirected edges influences the self-reduction step ($O(n^3)$), though the overall impact is small (as seen in Figures 8, 9, and 10).
>
> Regarding suggestions:
>
> - Thank you, we will correct this.
>
> -  We adopted this convention from previous papers on the topic. It is convenient because nodes can be used directly as indices. However, we understand the potential confusion and will add a clarifying remark.
>
> - We are using equations (3) and (4) on page 4 in the proof of Lemma 1, which are exactly your calculations. We should have mentioned this explicitly — thank you for pointing it out.
> Regarding Proposition 2: Let $n$ be the number of different prime factors. Since the smallest possible prime factor is $2$ and $z$ is the product of all its prime factors, $2^n \le z$, which gives the bound. We will add this to the proof.
>
> - Thank you, we will correct this.
>
> - This behavior can occur because matrix entries may be negative. For example, consider two disjoint paths between nodes $i$ and $j$ of the same length, where all weights are equal except for the last edges, which are negatives of each other. With a bit more care, one can construct such examples that stem from tree-shaped SCMs. We will add an example to the appendix of our paper.

---

> > ### Comment · Reviewer_hSv9 · 2025-08-03
> >
> > I thank the reviewer for the detailed response. I have also read through the responses to other reviews. I am happy that the authors are willing to accept the suggestions and provide important clarifications to help me understand the technical strengths of the paper. I will raise my score.

---

### Official Review · Reviewer_KPLf · 2025-06-25

**Clarity:** 2
**Significance:** 3
**Originality:** 2
**Rating:** 4
**Confidence:** 2

**Summary:**

This paper focuses on the problem of generic identifiability in linear structural equation models (SEMs) with tree-structured graphs. The authors build on the recent complete algorithm by Gupta and Bläser, which determines the identifiability of model parameters with a high computational cost of $O(n^6)$.
To improve efficiency, this paper proposes a series of optimizations for each step in the Gupta-Bläser framework. These include simplifying the covariance computations using Wright's trek rule, applying generalized Schwartz–Zippel-based identity testing to replace symbolic rank checks, using modular arithmetic to avoid large integer computations, and designing an efficient graph-based method for finding identifying cycles and paths. Together, these steps reduce the overall runtime to $O(n^3 \log^2 n)$, while preserving correctness guarantees with high probability.
The method is implemented and evaluated empirically on synthetic SEM instances, showing strong improvements in runtime and coverage compared to existing tools.

**Questions:**

(1) Can the authors give a realistic example where the data distribution would naturally follow a tree-structured linear SEM?

(2) If the given tree structure contains some errors, such as incorrect edge directions or wrong connections, what kind of results would the algorithm produce?

**Ethical Concerns:**

["NO or VERY MINOR ethics concerns only"]

**Final Justification:**

While the focus on tree-shaped structures is somewhat too specific, it nonetheless represents a meaningful contribution to the field. After the discussion, I maintain a generally positive evaluation.

**Limitations:**

See weakness and questions.

**Paper Formatting Concerns:**

no major formatting issues in this paper

**Quality:**

3

**Strengths And Weaknesses:**

Strengths:

(1) The reduction in computational complexity brought by the optimized algorithm makes it more practical for moderate-sized SEMs.

(2) The proposed optimizations still maintain correctness with high probability under a randomized setting.

(3) The method comes with a C++ implementation, which further improves its runtime efficiency.

Weakness

(1) The method only applies to tree-structured linear SEMs, which is too restrictive and therefore limits its applicability in practice.

(2) The specific optimizations, such as using the Generalized Schwartz–Zippel lemma for randomized polynomial identity testing, are based on standard existing techniques.

(3) The method is evaluated on only one dataset.

---

> ### Author Rebuttal · Authors · 2025-07-30
>
> Thank you for your thoughtful comments.
>
> Regarding the weaknesses:
>
> (1) It is true that tree-shaped SCMs represent a restricted class. However, the complexity of generic identification has remained an open problem for more than 20 years. In light of the (related) hardness results by Dörfler et al. (2024), the problem may indeed be computationally difficult. Therefore, it is reasonable to search either for criteria that are efficient but not complete, or for classes of SCMs for which the identification problem can be solved. We here follow the second path, as has also been done by van der Zander et al. (2022) or Gupta and Bläser.
>
> (2) While it is true that some of the underlying techniques—such as the Schwartz-Zippel lemma or self-reduction—are well-known in algorithm theory, they are rarely used in practical implementations. Translating the algorithm by Gupta and Bläser into practice was a highly nontrivial task. In particular, the idea of choosing random values only once at the beginning is novel, and care must be taken with dependencies, especially in the self-reduction part, which required new local update techniques, too. We improve upon the algorithm by Gupta and Bläser by a factor of more than $n^2$ (nearly $n^3$ for large $n$), which is crucial for making it practical. It saves a factor of about $10.000$ for SCMs of size $n = 100$. (More details on this can be found in the answer to reviewer 8jdN).
>
> (3) While it is true that we only used one graph per input size to draw the figures for simplicity, the running time of our algorithm does only depend on two parameters of the graph: whether there are missing edges to the root and the length of an identifying cycle. In the first case identification is very fast. In the second case, the most costly part of the algorithm uses linear algebra techniques (iterated matrix products) to find the identifying cycles and is oblivious to the actual edges in the SCM, since the product matrices get dense very fast. The longer the cycle, the more time we need, since we have to compute longer matrix products. However, this influence is also small, since we use binary search and always need to test for length $n/2$. If the cycle is long ($n$ in the worst case), then we always branch in the more costly part in the search. But the overall effect is small.  There are some parts in the algorithm where the actual edges play some role (computation of covariances, self reduction), but they are of lower complexity and their influence is minor (as can be seen in Figures 7,8,9).
>
> Regarding the questions:
>
> (1) Finding efficient criteria for determining whether a graph is generically identifiable has been a long-standing open problem—see, for example, Foygel et al. (2012, Problem 1), which remains unresolved. (Full references provided in our paper.) Consequently, investigating subclasses of SCMs for which identifiability can be determined is a natural approach. Moreover, in cases involving hierarchical data, tree-shaped SCMs arise naturally—see, for instance, Thorson et al. (2023). Of course, in practice, the tree structure may not be perfect. In such cases, one could attempt to remove a few edges to obtain a tree. If this resulting SCM is not generically identifiable, then the original one will not be either. The converse, however, does not hold, and in such cases, the identification result must be manually transferred to the original instance.
>
> (2) Generic identifiability is a property of an SCM, not of a particular data set. It means that all parameters are uniquely determined for almost all sets of observed covariances. (This is why, for example, Brito and Pearl (2002) refer to it as "almost everywhere identifiability.") Typically, one also obtains circuits that compute these parameters. So the "easy" answer to your question is that our algorithm will determine whether the perturbed SCM is generically identifiable, and if so, it will produce expressions for the parameters. More interesting, of course, is the question you intended: whether we can transfer the results from the perturbed SCM to the original one. Since we still lack a clear understanding of the complexity of generic identification, we are far from having general methods to determine how modifying a few edges affects the number of solutions. However, depending on the perturbations, certain properties might be preserved. For instance, adding directed edges will preserve nonidentifiability.

---

> > ### Comment · Reviewer_KPLf · 2025-08-04
> >
> > Thanks for your response, my questions have been addressed. Overall, while the tree-shaped structure is a fairly specific assumption in practice, I understand it as a reasonable trade-off, and improving the identification speed under this setting is still valuable. Therefore, I will maintain my current score.

---

### Official Review · Reviewer_wxcS · 2025-07-01

**Clarity:** 3
**Significance:** 2
**Originality:** 2
**Rating:** 4
**Confidence:** 3

**Summary:**

This paper provides an algorithm to identify linear tree-shaped SCMs. The contribution of this paper lies in the O((n^3)*(log^2(n))) temporal complexity of the proposed algorithm compared to the state of the art O(n^6).

**Questions:**

* An overview of the algorithm's steps and their respective complexity would be useful.

* In my opinion, it is very important to give more intuition as to why this very specific class of linear tree-shaped SCMs are interesting and useful.

* The benchmark section should be extended.
For exemple, how do your algorithm perform when the size of the cycles of missing bidirected edges vary.
You explain that treeID performs especially bad because of the implementation of the identity test. Could you show how treeID with a better identity test implementation perform?
The plotted graphs should also include the theoretical complexities of the different algorithms.
What is the goal of adding the black plot which shows a worst version of your algorithm (with O(n^4) detection of identifying cycles)? I do not understand the added value.
For figure 5: You plot the execution time of SEMID HTC while saying that it cannot identify any of the cases. I do not understand the added value. I believe it would be more relevant to compare with another algorithm that is complete (e.g. the ID algorithm) ideally the most efficient one .

* I believe "Gupta and Bläser 2024a" is not cited properly in the abstract.

**Ethical Concerns:**

["NO or VERY MINOR ethics concerns only"]

**Final Justification:**

The authors addressed most of my questions so I slightly increased my score. However, I remain somewhat unconvinced of its broad applicability

**Limitations:**

Yes

**Paper Formatting Concerns:**

No concerns

**Quality:**

2

**Strengths And Weaknesses:**

The main strength of this paper lies in the thorough step by step description of the algorithm.
Moreover, the improvement of temporal complexity allows the practical identification of SCMs with more nodes. The authors are clear about their contributions do not over sell it. While the improvement described by the authors seem sound, all proofs are left to the appendix.
The biggest weakness of this paper in my opinion is the small significance and originality and especially the lack of motivation. The authors focus on identifying a very restrictive type of SCMs (linear and tree-shaped), however they do not give any example of application where this specific task is useful. The only argument in favor of this line of study is the fact that identifying a broader class of SCMs would be harder.
On top of that, the author do not give the first algorithm to solve this problem nor even the first polynomial algorithm. They only improve an existing algorithm.
Lastly, the benchmarking section seems fairly weak when the only contribution is an improved theoretical temporal complexity.

---

> ### Author Rebuttal · Authors · 2025-07-30
>
> Thank you for your thoughtful comments.
>
> Regarding weaknesses:
>
> - The complexity of generic identifiability is a long-standing open problem: This means one can either look for methods that are not complete or for restricted graph classes for which we can design an efficient complete algorithm. This work takes the second route as have done others before, like van der Zander et al. (2022) or Gupta and Bläser (2024), all references are in the paper.
>
> - While the algorithm by Gupta and Bläser is also polynomial time, our algorithm is the first that is practical. We are not aware of an implementation of the algorithm by Gupta and Bläser. With some extra effort, we could have also implemented it using our code framework. However, as already pointed out in the answer to reviewer 8jdN, the algorithm is not practical even for moderate $n$: The running time of the Gupta and Bläser algorithm is $O(n^6)$, while ours is $O(n^3 \log^2 n)$.  Since the Gupta and Bläser algorithm uses similar techniques to ours, the constants hidden in the $O$-notation will be comparable. Therefore, our implementation is faster by a factor of at least $n^2$ on moderately sized instances, which for instance means a factor of $10000$ for $n = 100$.
>
> Regarding questions  (our answer items follow the same order as the question items):
>
> - Thank you for providing this nice idea. Page 5 contains an outline of the algorithm at the top, but we will add a more detailed description with complexities.
>
> - Finding efficient criteria for deciding whether an SCM is generically identifiable has been a long-standing open problem, see e.g. Foygel et al. (2012, Problem 1). Therefore, looking for subclasses of SCMs for which this is possible is a natural approach to this problem. Moreover, whenever we have hierarchical data, tree-shaped SCMs naturally occur, see for instance Thorson et al. (2023).
>
> - Regarding benchmarking: We are happy to extend the benchmarking. However, the running time of our algorithm is very stable, it essentially depends on only two parameters: whether there are missing edges to the root, and the length of the identifying cycles. Beyond that, the specific graph structure has little impact. If there are missing edges to the root, we can identify almost immediately (as shown in Figures 6 and 7). After that, the runtime depends only on the length of the missing cycle, since we use iterated matrix products to search for all potential missing cycles in parallel. However, since we use binary search to find the length of the missing cycle (which is the most costly step), this also has little influence, since we always need to check for length $n/2$. If the length is $n$, then we will always branch in the more costly part during binary search, so this is the worst case.  Similar observations apply to the runtime of the treeID algorithm, aside from the inefficiencies in identity testing, and HTC.
> Regarding your concrete questions:
>   - For exemple, how do your algorithm perform when the size of the cycles of missing bidirected edges vary:
> In Figure 5 we use the worst case for our algorithm, only one missing cycle which contains all nodes. In the random graphs in Figures 8,9,10, the identifying cycles are short, since this happens with high probability in a random graph (see also Section D.2). These are the two extreme cases. One could add cases in between, these would however be quite comparable.
>   - You explain that treeID performs especially bad because of the implementation of the identity test. Could you show how treeID with a better identity test implementation perform: This would require recoding treeID, since we think, after we inspected the code, that the current implementation uses a computer algebra system to perform the identity test, which would be hard to remove. Even with more efficient identity tests, the treeID algorithm still brute-forces over all potential cycles, which can be exponentially many.
>   - The plotted graphs should also include the theoretical complexities of the different algorithms: One can do this, however, this would mean that one has to set the constant of the O-notation to some "guessed" value to fit the observed running times. But we agree that this is helpful for comparison reasons and we will add them.
>   - What is the goal of adding the black plot which shows a worst version of your algorithm (with O(n^4) detection of identifying cycles)? I do not understand the added value: While $O(n^3 \log^2n)$ is asymptotically faster than $O(n^4)$, it is not clear whether this is true for small $n$ (due to hidden constants). (See lines 965-966 in the paper.) The plot confirms that the break-even point is indeed small.
>   - For figure 5: You plot the execution time of SEMID HTC while saying that it cannot identify any of the cases. I do not understand the added value: This still tells you something about the running time of HTC, even if it fails to identify any parameters. In this case, it will perform $n$ max-cut computations, which is a lower bound for its running time in general, since it performs between $n$ and $n^2$ many max-cut computations depending on the instance.
>   - I believe it would be more relevant to compare with another algorithm that is complete (e.g. the ID algorithm) ideally the most efficient one: There are only two complete methods known. One is based on Gröbner bases. This only works for small graphs. For instance, van der Zander et al. (2022) report that the Gröbner basis algorithm took more than four months (sic!) on the benchmark of Figure 4 (where we need 8s). The second one is based on the existential theory of the reals, see Dörfler et al. (2024). We are not aware of any implementation, but it will, like the Gröbner basis approach, only work for small $n$. So there are no efficient complete algorithms known for general graphs. In the light of the related hardness results by Dörfler et al. (2024), the problem might be hard in general. We do not know what you mean by "the ID algorithm". The most general algorithm we know is the ACID algorithm by Kumor et al. (2020). However, on tree-shaped SCMs it has the same problems as HTC, as shown by van der Zander et al. (2022), see lines 1180 - 1184. As ACID is an extension of HTC, it is more involved and slower.
>
> - Thank you for pointing this out, we will add the bibliographic details of the reference to the abstract.

---

> > ### Comment · Reviewer_wxcS · 2025-08-02
> >
> > Thank you for your very detailed and thoughtful rebuttal. Your clarifications on the algorithm’s complexity, implementation, and empirical evaluation are appreciated, and your comparisons with prior work are informative. That said, I remain only partially convinced on a few points: by the ID algorithm, I meant the work presented in "Complete Identification Methods for the Causal Hierarchy" by Shpitser and Pearl (2008), where they introduce the Y-rooted C-tree and prove that it completely characterizes the non-identifiability of the direct effect in a nonparametric setting. While I understand that your focus is on generic identifiability in linear SCMs, I think it would be helpful to clearly position your notion of identifiability in relation to the broader causal inference literature, especially to avoid confusion with the well-established ID framework for causal effect identifiability. Regarding the relevance of the "linear tree-shaped SCM" setting: I remain somewhat unconvinced of its broad applicability, but I acknowledge your motivating examples and understand why this subclass is of interest. Taking into account the overall quality of the paper and the other positive reviews, I will reconsider my score.

---

> > > ### Author Response · Authors · 2025-08-03
> > >
> > > Thank you very much for your feedback. Now we understand what you mean. The ID algorithm by Shpitser and Pearl is designed for nonparametric identification, in this sense the problem is different. Pearl [Causality, 2nd ed., Cambridge, 2009, page 154, beginning of Section 5.3.2] writes about parametric identification: "The identification results of the previous section [for parametric models] are significantly more powerful than those obtained in Chapters 3 and 4 for nonparametric models [which treat the ID algorithm]." The generic identification of general graphs will most likely need solving polynomial equations, which the ID algorithm cannot do. That being said, in the above quote, Pearls goes on with "Nonparametric models should nevertheless be studied by parametric modelers [...]". So we agree that we should discuss our results in a broader context.

---

### Official Review · Reviewer_8jdN · 2025-07-02

**Clarity:** 3
**Significance:** 4
**Originality:** 4
**Rating:** 5
**Confidence:** 4

**Summary:**

In linear structural causal models (SCMs), where the directed edges form a tree and undirected edges are arbitrary, Gupta and Blaeser (AAAI 2024, https://arxiv.org/abs/2311.14058) have demonstrated a polynomial-time algorithm for deciding generic identifiability. The present paper simplifies and implements the algorithm of Gupta and Blaeser. In particular, it uses the DeMillo–Lipton–Schwartz–Zippel algorithm for polynomial identity testing (https://www.sciencedirect.com/science/article/abs/pii/0020019078900674), and several further improvements in cycle-detection.

**Questions:**

Q1: To push the practical relevance further, one could ask whether in data that are not generated by tree-shaped SCM, but by some SCM of low structural complexity, the method performs better than commonly considered heuristics for the general, non-tree-shaped case. I would imagine it may improve upon their results substantially?

**Ethical Concerns:**

["NO or VERY MINOR ethics concerns only"]

**Final Justification:**

I really like the paper. While perhaps limited in its practical importance, the contribution is very elegant and completes the picture of complexity of learning causal models. The numerical results are very strong, considering the focus has been on the theory.

**Limitations:**

The limitations are well discussed.

**Paper Formatting Concerns:**

No formatting concerns, other than the paper ending rather abruptly on page 9.

**Quality:**

4

**Strengths And Weaknesses:**

Strengths:

S1: The paper makes an substantial improvement on computational complexity of a fundamental problem, but also makes a substantial effort on implementing the approach in practice.

S2: Overall, the paper and esp. the use of DeMillo–Lipton–Schwartz–Zippel is very elegant, but also practically useful and well motivated.

Weaknesses:

W1: The paper follows the same steps as Gupta and Blaeser (AAAI 2024, https://arxiv.org/abs/2311.14058), except better. Both intrinsically and in terms of the writeup, there is some overlap. Having said that, either it is self-plagiarism or "the greatest form of flattery", so I doubt that Gupta and Blaeser would mind.

W2: While the writeup is generally to be commended, further improvements could be made. Starting from Section 4.2, the paper becomes rather technical, less well organized, and the 9-page version skips several important definitions (e.g., a trek, Word-RAM, ring homomorphisms). It may be worth skipping some of the standard definitions (such as SCMs), perhaps, and including those less common in AI/ML venues instead? Also, the papers ends in Section 5.3 very abruptly.

---

> ### Author Rebuttal · Authors · 2025-07-30
>
> Thank you for your thoughtful comments.
>
> Regarding the weaknesses:
>
> W1: While we generally follow the strategy of Gupta and Bläser, we would like to emphasize that our improvements are nontrivial and crucial for making the algorithm practical. Moreover, one can interpret the soundness proof by Gupta and Bläser as suggesting that identifying cycles is, in some sense, the only viable approach to solving the problem.
>
> The running time of the Gupta and Bläser algorithm is $O(n^6)$, whereas ours is $O(n^3 \log^2 n)$. Of course, the additional polylogarithmic factor in our bound is non-negligible for small and moderate values of $n$. However, as shown in Figure 5, for $n = 100$, our algorithm is already faster than $O(n^4)$ (as mentioned in the paper, the break-even point is around $n = 50$). Since the Gupta and Bläser algorithm employs techniques similar to ours, the constants hidden in the $O$-notation of a potential implementation of the Gupta and Bläser algorithm will be comparable to ours. Therefore, our implementation is faster by a factor of at least $n^2$ on moderately sized instances — for example, a factor of $10.000$ (!) for $n = 100$.
>
> W2: We appreciate your comments regarding the write-up. Since the paper lies at the intersection of algebraic statistics and causality and employs methods from algebraic algorithms and computer algebra, many concepts from different areas need to be introduced. Our intention was to first provide a gentle introduction to the topic, then explain our main ideas for the improvements in Sections 4.1 and 4.2 using the example of covariance computation, and finally present the further improvements in Section 4.3, with more details provided in the appendix. But we see your points and will incorporate your suggestions.
>
> Regarding the questions:
>
> Q1: Generic identifiability is a property of an SCM, not of a particular data set. Given a tree-shaped SCM, we produce circuits (or determine that none exist) that compute the parameters for almost all possible observed sets of covariances. This does not directly apply to the scenario outlined in your question. However, one could attempt to find a tree-shaped SCM in a first step that approximates the observed scenario, and then use our methods to estimate the parameters. This seems like an interesting direction for future research. Thank you for raising this question.

---

> > ### Comment · Reviewer_8jdN · 2025-08-01
> > **Thank you.**
> >
> > Thank you for your detailed response. I maintain my Accept score.

---

### Decision · Program_Chairs · 2025-09-17

**Decision:**

Accept (poster)

**Comment:**

The paper focuses on improving the computational complexity of identification for linear tree-shaped SCMs, which is an interesting and underxplored area in causality. It provides a few rigorous theoretical contributions and a sound analysis, and the reviewers have appreciated the elegance of its methods and the technical clarity. Some of the reviewers had concerns regarding to the limited experiments, but these have been adequately addressed in the rebuttal by the authors. On the other hand, some concerns remain about the method's broader applicability based on the quite restrictive assumptions (linear+tree shaped SCMs). Nevertheless, the paper's theoretical contributions are interesting enough to recommend an acceptance.